# Hydro-Geochemical Characteristics of the Shallow Alluvial Aquifer and Its Potential Artificial Recharge to Sustain the Low Flow of the Garonne River

Nazeer Asmael [1,*] , Alain Dupuy [1], Paul McLachlan [2] and Michel Franceschi [1]

1    ENSEGID-Bordeaux INP, EPOC UMR 5805, University of Bordeaux, CNRS, 1 allée Fernand Daguin, 33600 Pessac, France; alain.dupuy@ensegid.fr (A.D.); michel.franceschi@ensegid.fr (M.F.)
2    Department of Geoscience, Aarhus University, Nordre Ringgade 1, Aarhus C, 8000 Aarhus, Denmark; pm@geo.au.dk
*    Correspondence: nazeer.asmael@ensegid.fr

**Abstract:** The complex and interconnected water challenges linked to global climate change and natural and anthropogenic water resources pressure have become major challenges in the 21st century. The Garonne River and its accompanying alluvial aquifers are considered the most important source for agricultural activities in the Garonne Valley, Nouvelle-Aquitaine Region, southwest France. The water is used for irrigation in summer and to reduce frost damage in spring. The alluvial shallow aquifer is recharged by rainfall, lateral inflow from the hillside, and seepage from the riverbed during the flood periods. The aquifer maintains the flow of the river during dry periods. Moreover, the potential recharge of this aquifer is particularly sensitive to annual climatic fluctuations and consequently affects surrounding ecosystems and related socio-economic activities. The increasing impacts of climate change have increased the concern about the availability of these resources. Various adaptation strategies have been considered to mitigate and adapt to the new situation in southwest France. The artificial recharge of the alluvial aquifer is one such regional adaptation strategy to adapt to climate change. The study has two main objectives: to assess the natural and anthropogenic influence on the groundwater chemistry, and to model water infiltration, and understand the aquifer response and, consequently, the effects on river baseflow. The TAG (Technopole Agen-Garonne) project aims to increase the economic wealth of the region while respecting the region's agricultural traditions. Runoff water from the TAG zone is collected in retention basins and is a potential source to recharge the shallow alluvial aquifer. Sampling campaigns were carried out during the summer of 2019 to collect groundwater samples from several observation wells. Groundwater levels were measured in 132 wells/boreholes to determine the groundwater level fluctuations and create piezometric maps. Piper, spatial distribution, and ionic ratio plots were used to determine the dominant hydrochemical processes and to delineate the hydrochemical facies in the study area. The groundwater chemistry is controlled by silicate weathering and anthropogenic influence. Groundwater quality appears to be affected by the river water in the wells located in the low plain area. The measurements showed that the groundwater levels in the wells located near the river increase more than 2 m after a flood event. The artificial recharge has increased the groundwater level by more than 1 m close to the infiltration basin after a rainstorm. Similarly, a three-dimensional (3D) groundwater model shows a similar magnitude aquifer response to the induced infiltration. The modeling-obtained result shows that the infiltrated water would take about 4 months to reach the Garonne River, which is an appropriate time to maintain the river's low-flow and thermal buffering capacity, and thus the functioning of its ecosystems during dry periods.

**Keywords:** Garonne River; alluvial aquifer; climate change; water resources management; MAR (managed aquifer recharge); groundwater modeling; France

## 1. Introduction

The French alluvial aquifers, where most fertile agricultural lands and big cities are located, play an important role in groundwater use in France. Current and future research in hydrogeology and water resources cannot avoid the effects of climate change, especially on these shallow aquifers.

Located in the southwest of France, the Nouvelle-Aquitaine Region is characterized by rich history, landscapes, and economy, which results from mixing climate and geography. This vast sedimentary area at the foothills of the Pyrenees and the Massif Central Mountains is bordered by a vast sandy coastline, ending in the extreme south with a rocky coastline in the Basque area. It comes into contact with the sea only through the Gironde Estuary in the Arcachon Basin, the Adour Estuary, and the ports between the cities of Capbreton and Hendaye. This region is also considered the largest cultivated forest in Europe, a wine-growing area, with rich and varied agriculture and a huge variety of natural ecosystems [1]. In the Garonne Valley, located in the central part of the Nouvelle-Aquitaine Region where agricultural activities are well developed, irrigation allows crop production where water would otherwise be a limiting factor [2]. The Garonne alluvial aquifer is an important reservoir and is extensively used for agricultural purposes due to its wide extension, high productivity, and the ease and low cost of extraction. This aquifer generally maintains a direct hydraulic connection with the river, which drains or recharges it according to the season. It was formerly used as a drinking water source, but it has now been abandoned for this purpose due to its high vulnerability to pollution, particularly from agricultural activities. Several studies have outlined the possible negative impacts of climate change on the Garonne River discharge and the recharge (infiltration ratio) of its surrounding alluvial aquifers [1,3–8].

The interactions of groundwater with surface waters take place in a range of different settings [9,10]: (1) Rivers gain water from groundwater inflow through the streambed (gaining stream); (2) they lose water to groundwater by outflow through the streambed (losing stream); or (3) they do both, gaining water in some reaches and losing water in others, or gaining and losing water in the same reach at different times depending on groundwater and surface water levels as well as the permeability of riverbed and aquifers. The dynamic exchange scale between the Garonne River and its alluvial aquifer is identified as between 3 and 4 km wide and limited mainly to the low plain [11,12]. The groundwater naturally feeds the springs and watercourses in the Garonne Valley, particularly during low river level periods. This groundwater discharge depends mainly on winter and spring rainfall. Anthropogenic activities such as groundwater and surface water extraction also affect the hydraulic gradients between the aquifer and the river system. Since the 1960s, anthropogenic impacts on the river's flow regime have significantly increased, particularly irrigation practices that coincide with a low-flow period [13]. The decrease in the river discharges is generally related to low precipitation and flood magnitude, consequently reducing the water storage in the aquifers, and a decrease in snow and ice storage in the high Pyrenees Mountains [13,14]. The decline in river flows and groundwater levels will become greater as the pressures on water resources continue to grow due to human population growth and climate change impacts. These could impact water availability and temperature, impacting the related ecosystems [15,16]. According to [17], global warming in the southwest of France is likely to increase the frequency of heatwaves, the duration of summer droughts, and the number of violent storm-type events. It is also likely to decrease the duration, extent, and quantity of snow cover, summer precipitation, average summer discharges of rivers, and groundwater levels. Water resources can be considered a crucial issue to fight against climate change and adaptation measures that can be taken.

The objective is to slow the rainfall trajectory towards the sea, prevent runoff, and enhance infiltration. Taking part of the existing surface water to be recharged into the aquifers for later different use can be a pragmatic strategy for adapting to climate change and sustaining the low flow in the Garonne River. This economically interesting solution can improve the quantity and quality of the alluvial aquifer, which is significantly exposed to pollution. Furthermore, this solution can also assist local water management and allocation problems. The weathering

processes of the alluvium deposits play an important role in the groundwater quality of the unconfined alluvial aquifer. Hydrochemical data can be used to characterize this aquifer and specify different processes in the system and the effects of anthropogenic activities on the groundwater. The nitrate concentration of the alluvial aquifer is highly related to the hydrogeological context and surface land use. In the Garonne Valley, the alluvial aquifer is characterized by high nitrate concentration compared with the surface water [18]. The geological features of this aquifer and its shallow depth present a favorable condition that strongly affects its mineralization and vulnerability. In the Garonne Valley, almost 70% of water abstraction during low-water periods is used for irrigation [7]. Hence, the alluvial aquifer sustains extensive agricultural activities using fertilizers and a comprehensive irrigation system. It is degraded by nitrate pollution due to leaching from fertilizers in the return flow and consequently pollution is diffused into the groundwater system.

A large socio-economic project (TAG) of 240 ha in the study area has developed close to Agen, in southwest France. This project involves using large retention basins to collect runoff water for recharging the alluvial aquifer. This will allow reconstitution from the end of winter to the end of spring by using the alluvial aquifer for groundwater storage. Thus, it will increase the aquifer potential flow toward the Garonne River to sustain its low flow. Based on our measurements in August 2019, the river temperature was measured locally as 29 °C while the temperature of the drained groundwater into the river was 15 °C in the same period. Subsequently, the potential increase in drainage from the aquifer into the river during the dry period will ensure the functioning of its ecosystems. The shallow depth of the unconfined alluvial aquifer can be an advantage, allowing rapid and economical storage.

Hydrochemical data can be used to identify hydrogeological processes and understand the flow system behavior. Local-scale groundwater modeling can also be used to identify the response of the alluvial aquifer to local artificial recharge and to delineate the flow patterns. Hence, the main objectives of this paper are: (i) to assess the natural and anthropogenic influence on the groundwater chemistry and determine the main hydrochemical process controlling the groundwater chemistry; (ii) to model water infiltration, and understand the quantitative aquifer response to the artificial recharge and, consequently, the role of this methodology in maintaining river base flow; and (iii) to enhance the relationship between the hydro-system and the aquatic ecosystem along the river against climate change. This work will help to assess the potential of this experimental approach as a water management tool across the Garonne Valley as a nature-based, viable, and sustainable solution for extreme hydro-climatic events.

## 2. Description of the Study Area

The study site is located in the southwest of France, in the Lot-et-Garonne Department on the left side of the Garonne River near the city of Agen in the Nouvelle-Aquitaine Region (Figure 1). It covers an area of approximately 20 km$^2$. The Garonne Valley forms a vast plain, 5 to 8 km wide, bordered by asymmetrical hillsides that form an integral part of the valley. The hillsides on both sides are regularly cut by tributaries of the Garonne. The flat valley floor includes the river's major bed, structured by the alluvial terraces that form its limits. The average altitude of these terraces ranges between 10 and 15 m, allowing the buildings to be relatively protected against floods. The TAG area is a large development zone that is listed in strategic documents of the Lot-et-Garonne Department and meets an economic need in the Agen Agglomeration by preserving agricultural activities in this area according to the planned economic development and the allocated infrastructure. Small stream networks, ditches, marshes, old gravel pits, and small wetlands all contribute to the biodiversity of this area. The proximity of the Garonne River to this area constitutes its principal ecological connection. This economic project considers sustainable development by preserving considerable green spaces, collecting runoff water to be recharged into the shallow alluvial aquifer, using completely solar-powered lighting, and giving particular attention to the existing flora and fauna.

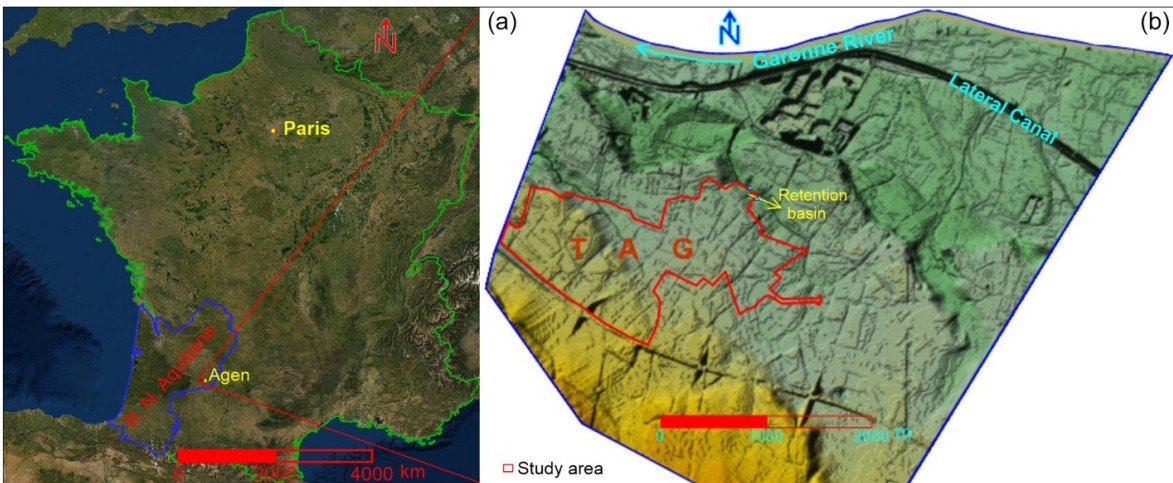

**Figure 1.** (**a**) Map of France, prepared by QGIS 3.16.12, free and open-source software, showing the location of the study area in Nouvelle-Aquitaine Region; (**b**) the topographic units of the study area according to IGN (French national geographic institute, https://geoservices.ign.fr/, accessed on 20 September 2021) spatial satellite data with a 5 m resolution (**b**) [19].

Three large retention basins have been created in the TAG area, with a total area of 0.8 ha. The catchment area of these basins varies between about 0.7 and 2 km², taking into consideration that this catchment area extends outside of project area. Run-off water from public spaces and existing houses in the project area is collected by lateral ditches along the roads and directed towards the retention basins. Runoff water from parcels greater than 1 hectare is collected by a small retention basin located on the parcel while the runoff water from parcels less than or equal to 1 ha is collected by the lateral ditches and drained to the large retention basins.

The study area is bordered to the south by hills, where the altitude is around 83 m, and to the north by the Garonne River, where the altitude is around 32 m. Three geographical units can be identified in the area (Figure 2). The first one occupies the southern and southwestern part with a steep slope, the second is located in the middle of the study area and characterized by moderate slopes, and the third extends to the north and northeast towards the Garonne River, where the slope remains gentle.

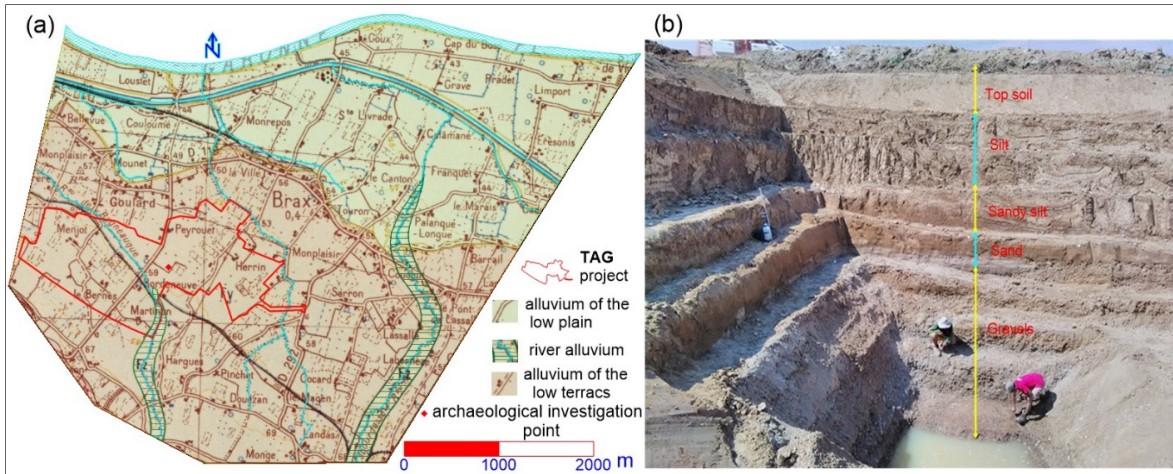

**Figure 2.** The different geological formation outcrop in the study area (**a**), based on [19,20]; a stratigraphic section of the alluvium of low terrace showing an archaeological investigation structure (**b**), based on [21].

From a geological point of view, at the end of the Mesozoic Era, tectonic movements began increasing until the middle of the Cenozoic Era. They have caused a gradual uplift of the Pyrenees Mountains, and deformations and faults in the ground. A succession of marine transgressions and regressions corresponding to climate changes has left an alternation of sedimentary rocks in the Nouvelle-Aquitaine Region. Less permeable clay and marl layers form a barrier to groundwater circulation.

The Garonne Valley is considered one of the largest alluvial plains in France [22]. The Garonne River sank into the Mesozoic deposits during the Quaternary and intensively deposited alluvium. The Quaternary alluvium is found along the Garonne River and its main tributaries. We can distinguish between current, recent, and ancient alluvial deposits, which form the major bed, the low plains, and the terraces, respectively [22]. The terraces were formed as a result of the successive excavation and infilling process in the valley. They are laid directly on the impermeable molassic bedrock. The ancient alluvial deposits are made up of silts, clays, and sandy gravel, and are represented by the low terrace in the study area. In contrast, the recent alluvium deposits, represented by the low plain in this area, consist of gravel and large pebbles intercalated with a sandy matrix. According to the geological map of France [20], the boundaries between different geological units are well observed in the study area (Figure 2). Different lithological formations have been distinguished during an archaeological study [21] conducted on the TAG area (Figure 2). The bottom of the structure was reached at a depth of 10 m, where the groundwater level was found to be 9.20 m on July 2018.

Figure 3 shows a geological cross section in the Garonne Valley between Aiguillon and Saint-Pierre-de-Buzet about 25 km downstream of the city of Agen, representing the vertical and horizontal extent of the alluvial deposits in the Garonne Valley.

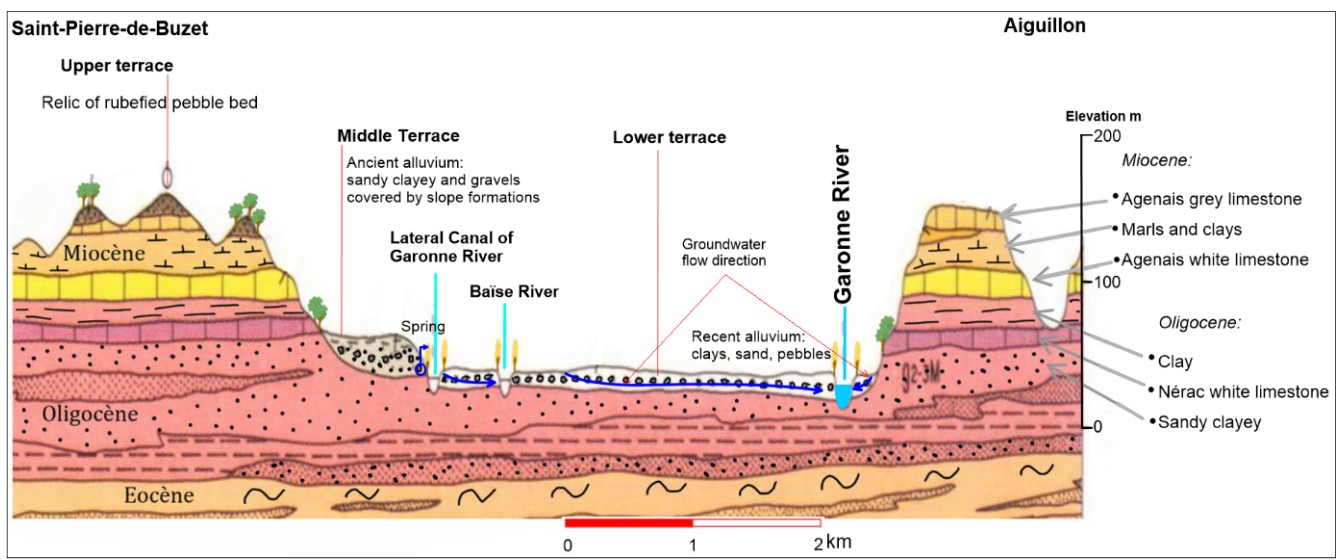

**Figure 3.** Geological cross-section in the Garonne Valley; from BRGM.

The Garonne River, with a drainage basin of 57,000 km$^2$ [23], is the third largest French Atlantic river in terms of discharge. The French and Spanish Pyrenees Mountains, the foothills of the Massif Central Mountains, and the plain between them feed the river. The hydrological behavior of the river is quite variable. It reflects the influence of Mediterranean and oceanic weather conditions and the effects of different landscapes and associated altitudes and slopes [24,25]. Hence, its flow is always influenced by the melting of the Aneto glacier, located in the Spanish Garonne, snow, rain, and drainage of the accompanying shallow alluvial aquifer. The Tonneins gauging station is about 40 km downstream from the city of Agen, and it is the lowest station not influenced by the tidal action of the ocean. The minimum objective discharge depends on station location; for Tonneins station, which

is about 8 km after the junction point between the Garonne River and its tributary, the Lot River, this discharge is 110 m$^3$/s. According to the data measured in Tonneins station between 1989 and 2022, the mean annual discharge rate of the river is 517 m$^3$s$^{-1}$. The lowest value measured in this period was 34.1 m$^3$s$^{-1}$ in August 1989, and the highest was 6190 m$^3$s$^{-1}$ during the floods of February 2021 (https://www.hydro.eaufrance.fr, accessed on 17 October 2022).

The river is usually flooded in winter or spring after the snow melts. This flood is sometimes followed by a significant low discharge in summer. According to [3], the possible effect of climate change on the river is as follows: an average 11% river discharge decrease during the low-flow period (July–October) and an increase in winter discharge followed by a decrease during the spring. A program to supply water to sustain its low flows and to satisfy the users' needs and the aquatic environment was initiated by the SMEAG (Syndicat Mixte d'Études et d'Aménagement de la Garonne) organization about 30 years ago using 12 reservoirs located in the mountain area. Around 70 Mm$^3$/year, mainly from hydroelectric reservoirs in the Pyrenees Mountains, are mobilized for this purpose. This supply can represent up to 50% of the river's discharge from mid-August to mid-September. In 2022, the Garonne River had the longest low-flow period in its recent history, which lasted almost four months continuously. Figure 4 shows the dates on which the target low flow was exceeded between 1960 and 2022, according to the data measured at the Tonneins station. This figure shows that the river appears to have an extremely low flow quite early since 1984 (the earliest one took place on 6 June 2006), and thus, the number of years in which the objective low flow has not been exceeded has decreased.

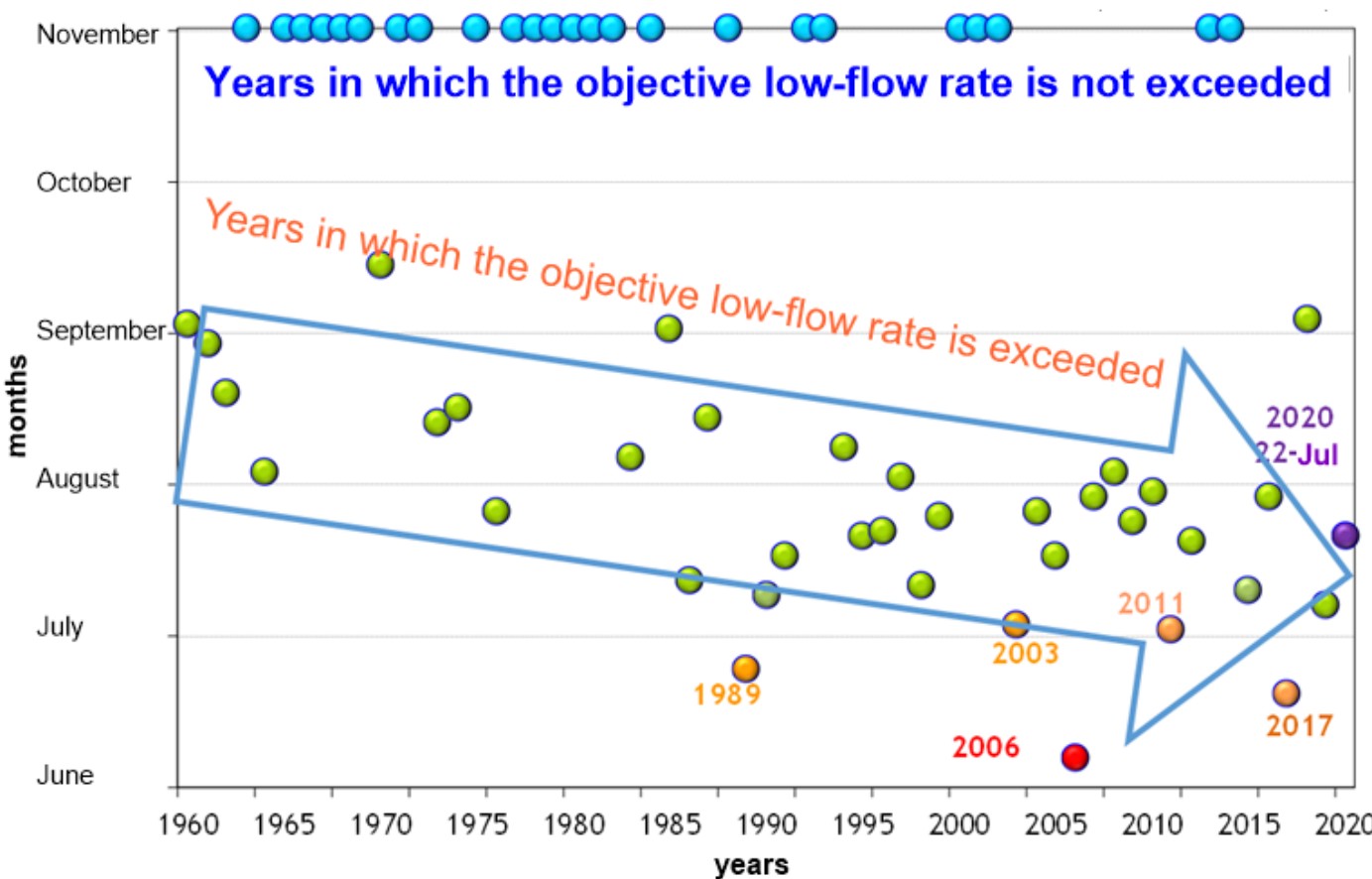

**Figure 4.** Low-flow index of the Garonne River at the Tonneins station; data source: SMEAG (https://www.smeag.fr, accessed on 20 September 2021), red and orange colours indicate that the river begins to experience low flow conditions early in the corresponding years.

The alluvial aquifer in the Garonne Valley is integrated into the river system. This aquifer plays an important role in water storage and hydraulic connection as a transmission zone between itself and the adjacent hillsides on one side and the Garonne River on the other side. The thickness of the aquifer varies from 4 to 10 m covered by a sandy-clayey and silty layer of 2 to 7 m. This aquifer tends to be locally semi-confined depending on the thickness of the clay and silt cover. The ancient alluvial deposits have undergone much more weathering than the recent formations. Thus, they are characterized by more clay deposits than the recent ones, which reduces the permeability of the aquifer. Hence, the low plain of recent alluvial deposits constitutes an abundant unconfined aquifer in direct hydraulic connection with the river. It feeds the river and sustains its base flow, particularly in the summer period, and it is recharged by the river during flood periods, reducing the river's flood risk. Subsequently, the plains alluviums constitute a buffer zone controlling the exchanges between the alluvial aquifer and the river. The aquifer–river interface comprises a complex mosaic of surface and subsurface flows of varying lengths, depths, directions, and residence times [26,27].

Consequently, the flow direction between the river and the surrounding alluvial aquifer can be reversed according to groundwater and surface water gradient. Since the Garonne River flows directly on the substratum molassic, the seepage rate will be limited by the hydraulic conductivities of the river banks. The underground part of the alluvial aquifer is located at a depth that varies from 10 to 15 m. This portion of the aquifer corresponds to an impermeable molassic bedrock from the Oligocene which constitutes the bottom of the aquifer. Groundwater seepages into the riverbed have been observed at several sites in the study area during summer. The aquifer contribution to the river base flow differs between the wet and dry years. The exchange between the alluvial aquifer and the river is illustrated in Figure 5.

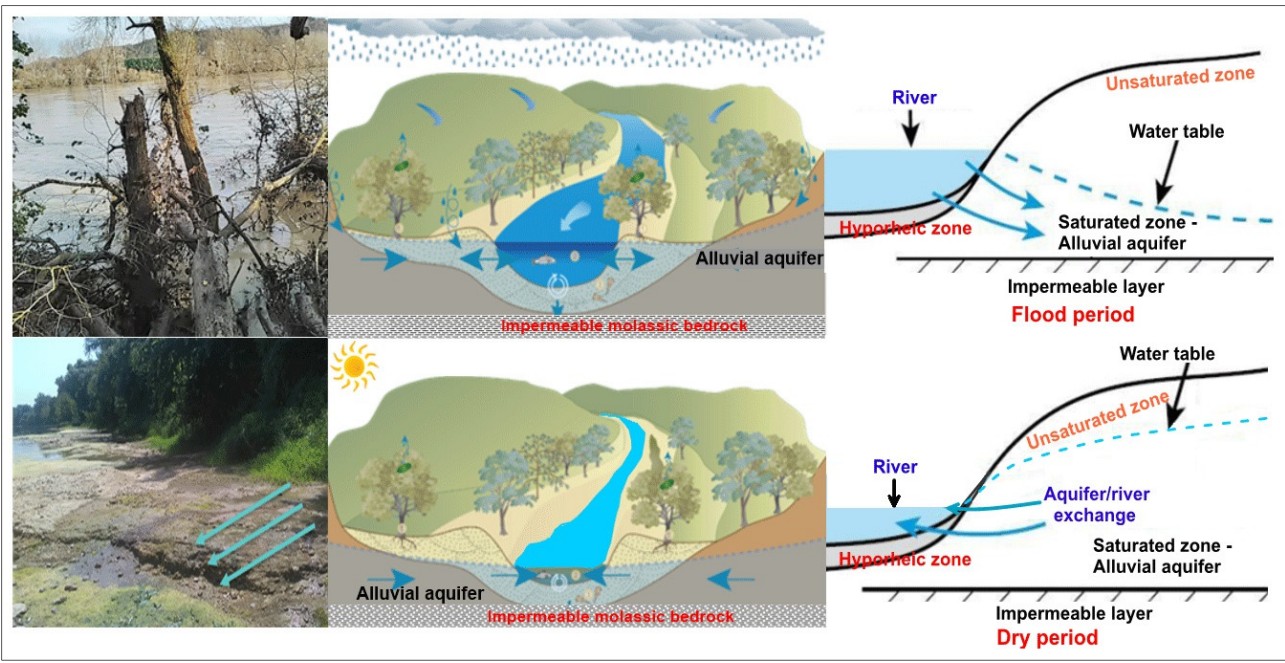

**Figure 5.** Alluvial aquifer/Garonne River interface and exchange patterns during flood and dry periods, based on [4,19,28].

## 3. The Concept of Artificial Groundwater Recharge in the TAG Area

The groundwater volume of an aquifer should be preserved, so the exploitation should only concern the yearly renewable volume of this aquifer. In the case of the shallow alluvial aquifer of the Garonne Valley, the quantitative imbalance could be caused by insufficient recharge of the aquifer by autumn or winter rainfall and is accentuated by significant

withdrawals in the summer period to meet the agricultural needs. Artificial groundwater recharge of this aquifer could be integrated into water resources management strategy to maintain long-term water sustainability and related ecosystems. Based on the observed hydraulic relationship between the Garonne River and the adjacent alluvial aquifer, the artificial recharge of this aquifer has been initiated to sustain the low flow of the river.

## 4. Materials and Methods

A total of 132 wells and boreholes, distributed across the study area (see Figures 6 and 7), were used to measure the groundwater levels. According to the owners of the wells, most of them reach the top of the molassic bedrock. The groundwater levels were measured monthly over 20 months (July 2018–February 2020) using a manual piezometer sensor. The piezometric map of March 2019 is shown in Figure 7. A total of 45 groundwater samples were collected during the fieldwork campaign in September 2018. Prior to sampling, each well was pre-pumped for 10 to 15 minutes to renew the groundwater. All samples were analyzed in the ENSEGID (Graduate School of Environmental, Geological, and Sustainable Development Engineering) Institute laboratory in Bordeaux, France. Nitrate measurement was carried out in the field using a HORIBA B-742 LAQUAtwin Nitrate meter. The pH and temperature were also measured in situ using a WTW MultiLine Multi 3630 IDS Multi-Parameter handheld box. The major ions of 45 samples were analyzed by ion chromatography using a Dionex ICS-900. $HCO_3^-$ was analyzed with a Metrohm 716 DM Titrimeter by titration with 0.1 N hydrochloric acid. The ionic mass balance of the samples shows that the majority of errors (75%) were lower than $+/-5\%$. The geochemical software PHREEQC (version 3.6.2.15100) [29] was used to calculate the saturation indices (S.I.) of the minerals present in the groundwater system.

Runoff water is collected in lateral ditches in the TAG area and directed to retention basins. This water is considered a potential source to recharge the shallow alluvial aquifer and its later use for preserved green areas in this industrial zone. The monitoring of groundwater levels in the study area over 20 months provided essential information on the hydrodynamic regime of the aquifer, the recharge/discharge zones, and the effects of abstraction and floods on the groundwater level.

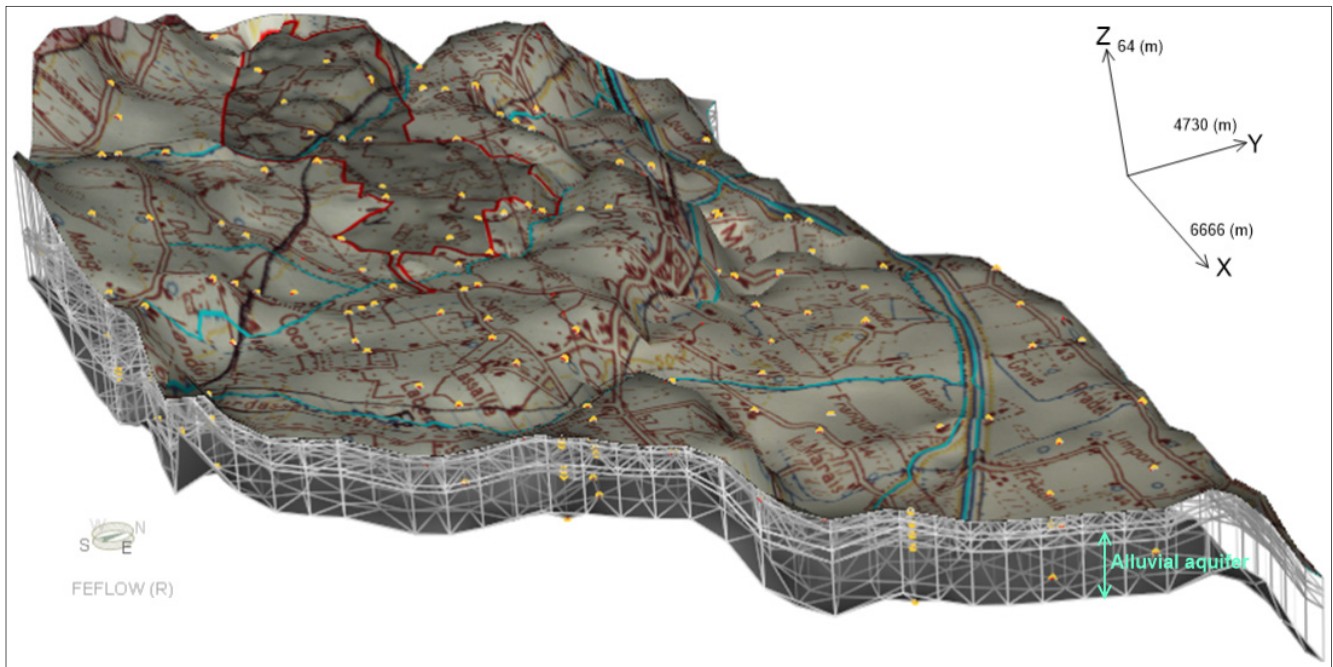

**Figure 6.** The three-dimensional (3D) representation model of the study area showing four layers and five slices, constituting the model domain [18].

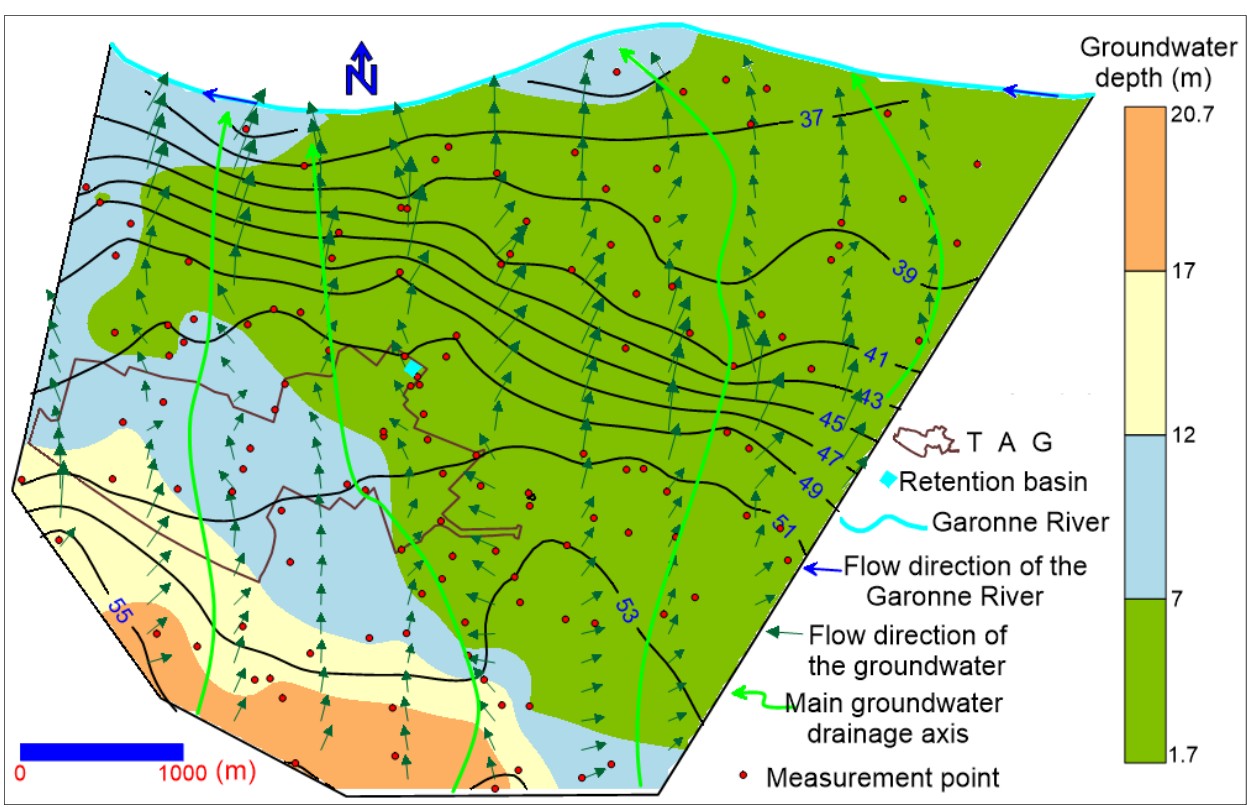

**Figure 7.** Location map of observation wells and the piezometric map showing groundwater flow direction and water table depth from the surface for March 2019.

The hydrodynamic model was developed to study the flow regime of the alluvial aquifer after implementing the artificial recharge system using an infiltration basin. Using FEFLOW 7.1 by DHI-WASY [30], a finite element subsurface flow and transport simulation system, the framework of super elements was defined, and then the 2D horizontal mesh of finite-element was generated using the automatic triangle option. The model top layer topography was determined based on IGN spatial satellite data with a 5 m resolution. Manual corrections concerning the auto-route bridges as well as the bridges of the lateral canal of the Garonne River located in the study area were made, respecting the topography around these bridges. After that, and based on the available information, three-dimensional (3D) slice elevation, layer properties, and boundary conditions were defined. Then, the advanced subsurface flow system was investigated under the steady state condition. The 3D model grid consists of four layers (Figure 6), and the thickness of each layer varies according to the topography and the geological formation. The first layer represents the topsoil, which is 1–3 m thick. The second layer consists of silt and sandy silt with a thickness varying between 1 and 7 m, increasing towards the hillslopes. The third layer represents a thin sandy layer of 1–2 m. The fourth layer consists of gravel, pebbles, and sandy clay housed in the molassic formations with a thickness varying between 6 and 10 m. The alluvial aquifer is represented by the third and fourth layers. QGIS (Geographic Information System) was used to visualize and pre-process the geographic data required for the model.

## 5. Result and Discussion

### 5.1. Piezometric Map Delineation and Groundwater Level Fluctuation

The piezometric map provides important information about the groundwater flow direction and the variation of the hydraulic gradient. The piezometric map for March 2019 is shown in Figure 7. This map shows that the piezometric levels vary between 55 m.a.s.l in the low terrace in the south and southwest of the study area, near the hillsides, and 35 m.a.s.l

in the northern part in the low plain near the Garonne River. The piezometric surface of the study area shows that there is hydraulic continuity between the two hydrogeological units of the low plain and low terrace. The hydraulic gradient is slightly marked in the southwest towards the hillsides, while it is more important between the low terrace and the low plain.

The significant variations in thickness and lithology of the alluvium lead to significant variations in the permeability of the alluvial aquifer in the study area. According to the groundwater levels measurement over 20 months, the northern part close to the river has shown more groundwater level fluctuation in both dry and flood periods according to the river level changes, and hence the exchange between the river and the aquifer. The groundwater flow direction is mainly oriented south–north towards the main drainage axis to the Garonne River. The hydraulic gradient is very variable from one sector to another of the study area, and it is 2.5 m·km$^{-1}$ in the south and 8.5 m·km$^{-1}$ in the southwest, where the isopiezes are parallel to the boundary, indicating a groundwater supply from the hillside. In the TAG area, the lowest value of hydraulic gradient (1.8 m·km$^{-1}$) was measured, indicating a high permeability of the aquifer. In the sector between the low terrace and the low plain, oriented southeast–northwest, where the isopiezes are generally equidistant, the highest hydraulic gradient value of about 11 m.km$^{-1}$ was measured. This value was measured at 3 m·km$^{-1}$ in the northeast of the study area, while it was measured to be 8 m·km$^{-1}$ to the northwest of this area. Groundwater depth changed between 1.7 and 20.7 m from the surface in the study area during March 2019.

Figure 8 shows groundwater level fluctuations at five measurement points in the study area. The groundwater levels measured in two boreholes located close to the retention basin show that the groundwater level increased between 1 and 0.5 m in the boreholes PZ1 and PZ6, respectively, after the rainfall event of November 2019 (Figure 8). The groundwater level measured in PZ10, located about 500 m upstream of the retention basin, showed a response due to natural input (Figure 8). Following the storm rainfall in the upstream part of the river's basin in November 2019, the Garonne River was flooded. The effect of this flood on the piezometric levels was measured in the two wells (FP84, FP92), located at 70 and 150 m from the river bank, respectively (Figure 8). These wells show a response important to this flood event as the groundwater levels increased by about 2 m in these wells. This observation shows a clear exchange between the Garonne River and its accompanying alluvial aquifer at the end of the dry period, where the river has well recharged this aquifer.

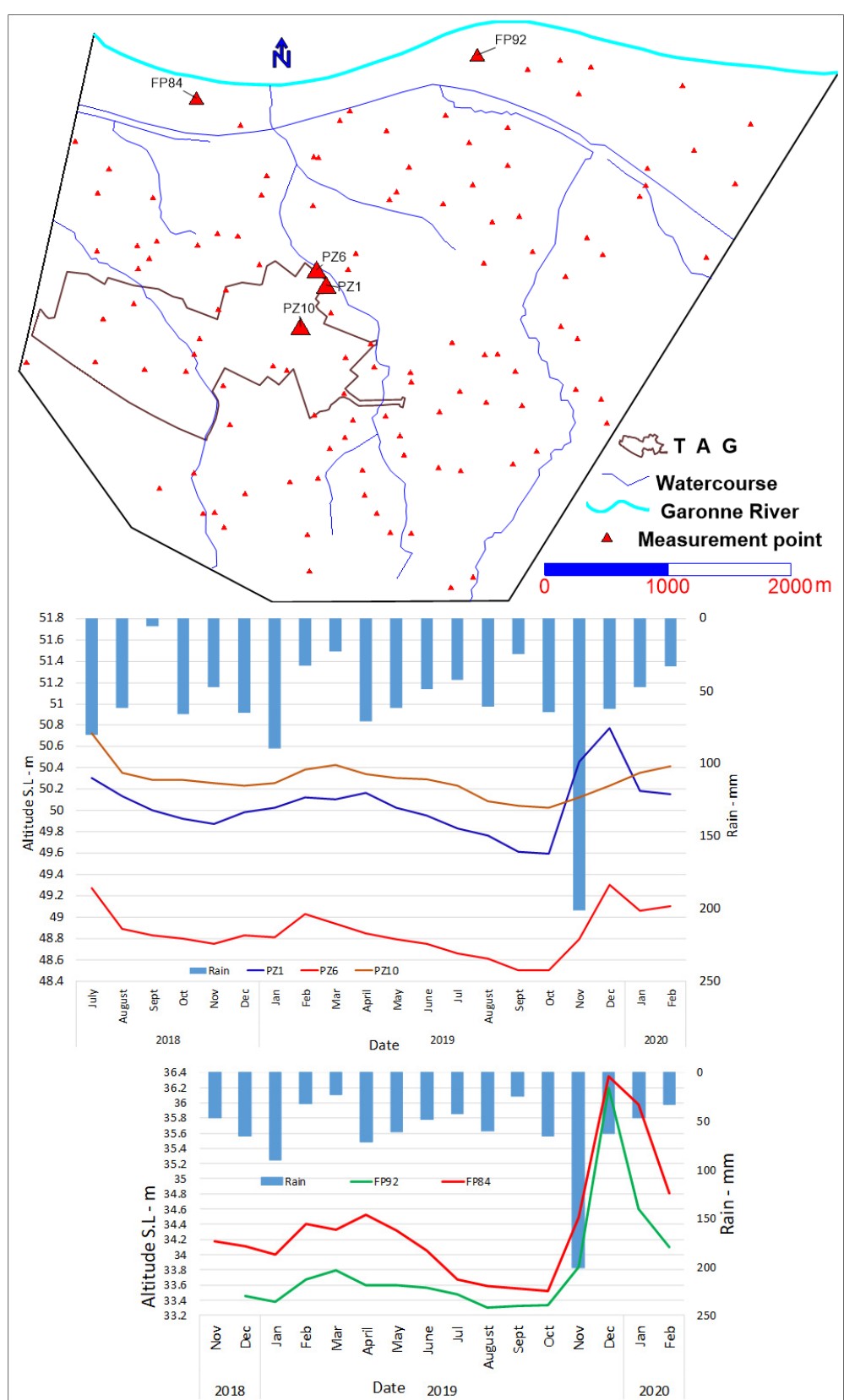

**Figure 8.** Monthly groundwater level fluctuations in two boreholes located close to the retention basin, one well about 500 m upstream of the infiltration basin, two wells close to the river, and monthly rainfall variation measured at Agen station close to the TAG area.

*5.2. Groundwater Hydrochemistry*

Different processes, such as groundwater flow, recharge and discharge, and water-rock reactions, control groundwater hydrochemistry. Along the groundwater flow direction, mineral weathering affects hydrochemistry during residence time [31,32]. Knowledge about groundwater quality is an important part of studying the hydrodynamics functioning of the aquifer.

5.2.1. Nitrates ($NO_3^-$)

The nitrate concentration was measured in situ in September 2018 in 50 wells (Figure 9). The measured values ranged between 12 and 117 mgL$^{-1}$. Additionally, more than two-thirds of the samples exceeded the drinking water threshold of 50 mgL$^{-1}$. Relatively high concentrations were measured in the central part of the study area. Nevertheless, nitrate concentrations were measured to be relatively low in the low plain of the Garonne River, where there is an exchange between the river and the aquifer. These low nitrate concentrations might have resulted from groundwater denitrification due to the input of dissolved organic carbon from the surface water or groundwater dilution by the Garonne River's water.

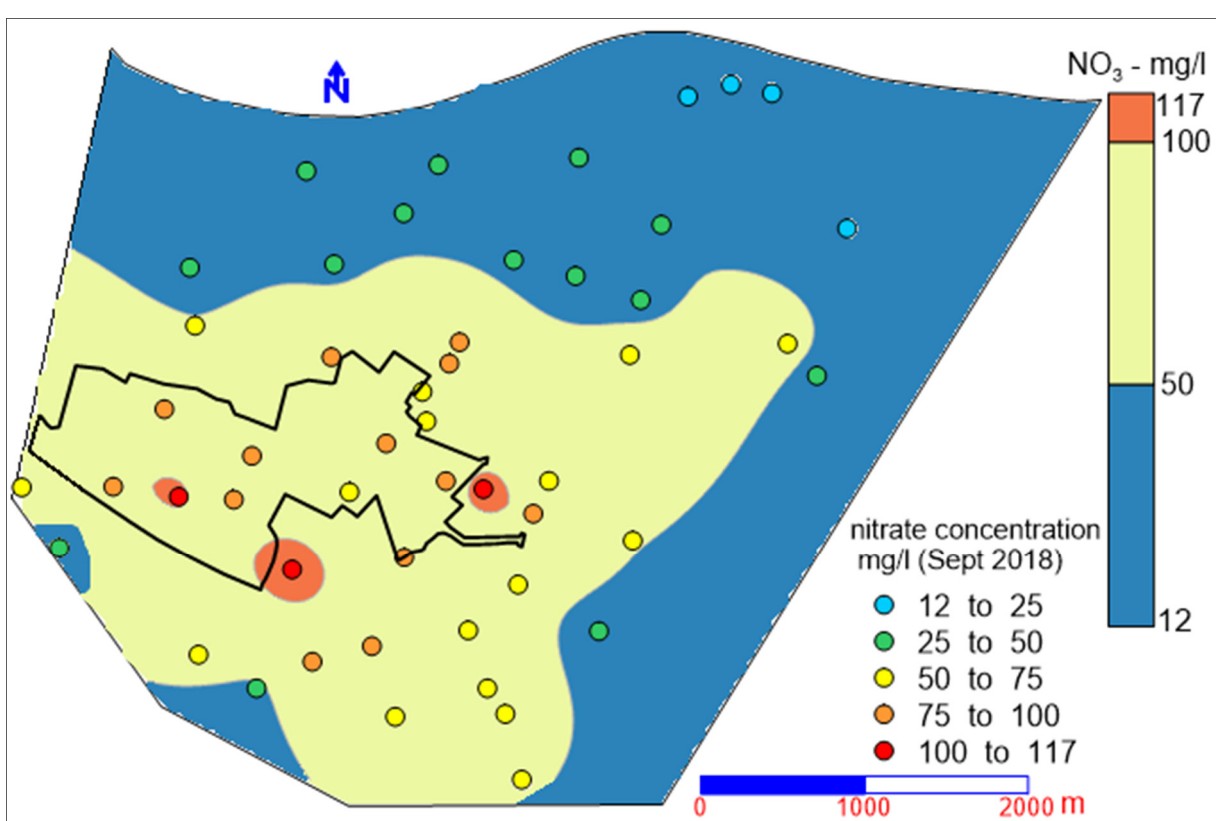

**Figure 9.** Spatial distribution of nitrate concentrations (zonation and points) measured in the alluvial aquifer of the study area, September 2018.

5.2.2. Dominated Hydrochemical Processes in the Alluvial Aquifer

To better understand the hydrogeological processes involved in the hydrochemical evolution of the aquifer system, groundwater mineral equilibrium calculations can be useful to predict the presence of reactive minerals and estimate the mineral reactivity of the groundwater system [33,34]. The saturation indices of certain minerals have been calculated and are presented in Table 1.

**Table 1.** Saturation index of minerals (anhydrite, carbonate, gypsum, halite, and quartz) in the alluvial aquifer calculated with PHREEQC.

| Mineral Saturation Index | Min | Max | Average | Standard Deviation |
|---|---|---|---|---|
| Anhydrite | −2.9 | −1.9 | −2.3 | 0.2 |
| Calcite | −1.3 | 0.1 | −0.6 | 0.3 |
| Dolomite | −2.8 | −0.5 | −1.7 | 0.5 |
| Gypsum | −2.5 | −1.5 | −1.9 | 0.2 |
| Halite | −8.5 | −6.6 | −7.5 | 0.4 |
| Quartz | −0.1 | 0.3 | 0.1 | 0.1 |

Calcium and magnesium concentrations have been plotted as a function of calculated $_pCO_2$ by PHREEQC, and the pure calcite and dolomite dissolution determined by PHREEQC (Figure 10).

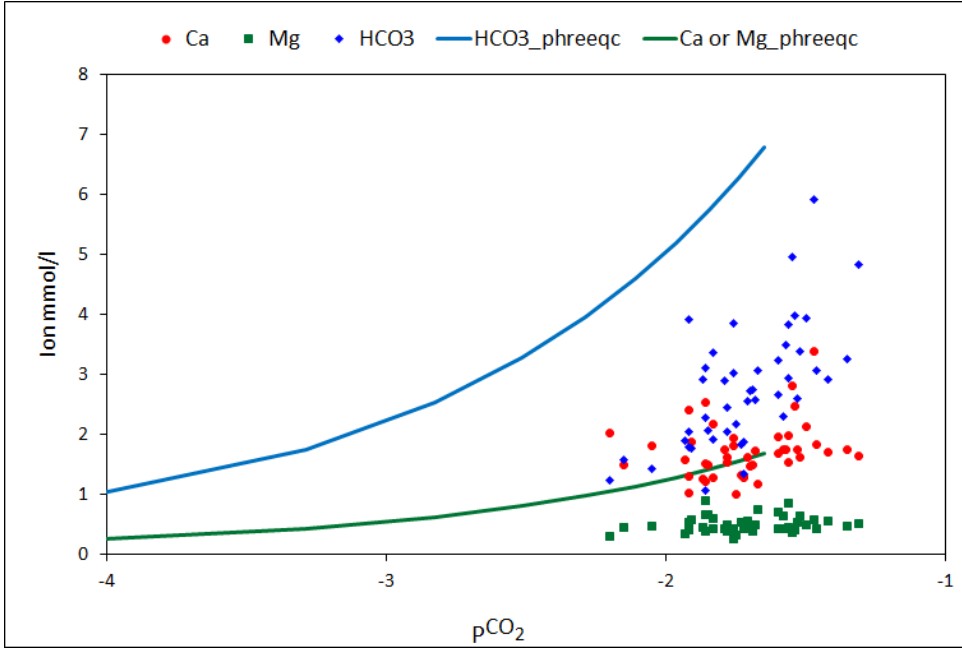

**Figure 10.** Calcium, magnesium, and alkalinity concentration in mmolL$^{-1}$ as a function of $_pCO_2$ (atm) calculated values; solid lines are the alkalinity, calcium, and magnesium determined with PHREEQC for pure calcite and dolomite dissolution.

Most water samples are near or below the PHREEQC line concerning the calcium, while all samples are below the PHREEQC lines for bicarbonate and magnesium (Figure 10). This means that the sources of calcium supply in groundwater, such as the dissolution of calcite, dolomite, and gypsum, are not the main process controlling groundwater chemistry in this case.

5.2.3. Hydrochemical Facies of the Groundwater

To better understand the hydrogeological processes involved in the aquifer system, the water type, according to Stuyfzand classification [35], is depicted together with a Piper diagram [36], as shown in Figure 11. The main water type and subtype have been determined for each water sample based on chloride, alkalinity, and the most important ions values. Concerning the chloride concentration, the sample can take the code H, S, b, B, f, F, g, and G with respect to increasing chloride concentration. In our case, groundwater is classified as oligohaline and soft (Cl changes between 10 and 88 mgL$^{-1}$), so the code varies between g and F. The g factor (relatively low chloride concentration) is mainly found in the low plain, where the aquifer is directly connected with the river, and the exchange

between them is well observed by the drainage during the dry periods and increasing the groundwater level during the flood periods. Regarding the alkalinity, the sample can take the code * or values from 1 to 9 corresponding to the increasing of alkalinity concentration. The result in our case shows that 7% of the samples were classified as slightly high, 71% as moderate, and 22% as low (alkalinity varying between 1 and 6 meqL$^{-1}$, the code changes between 1, 2, and 3, and the groundwater classifies as soft and very soft).

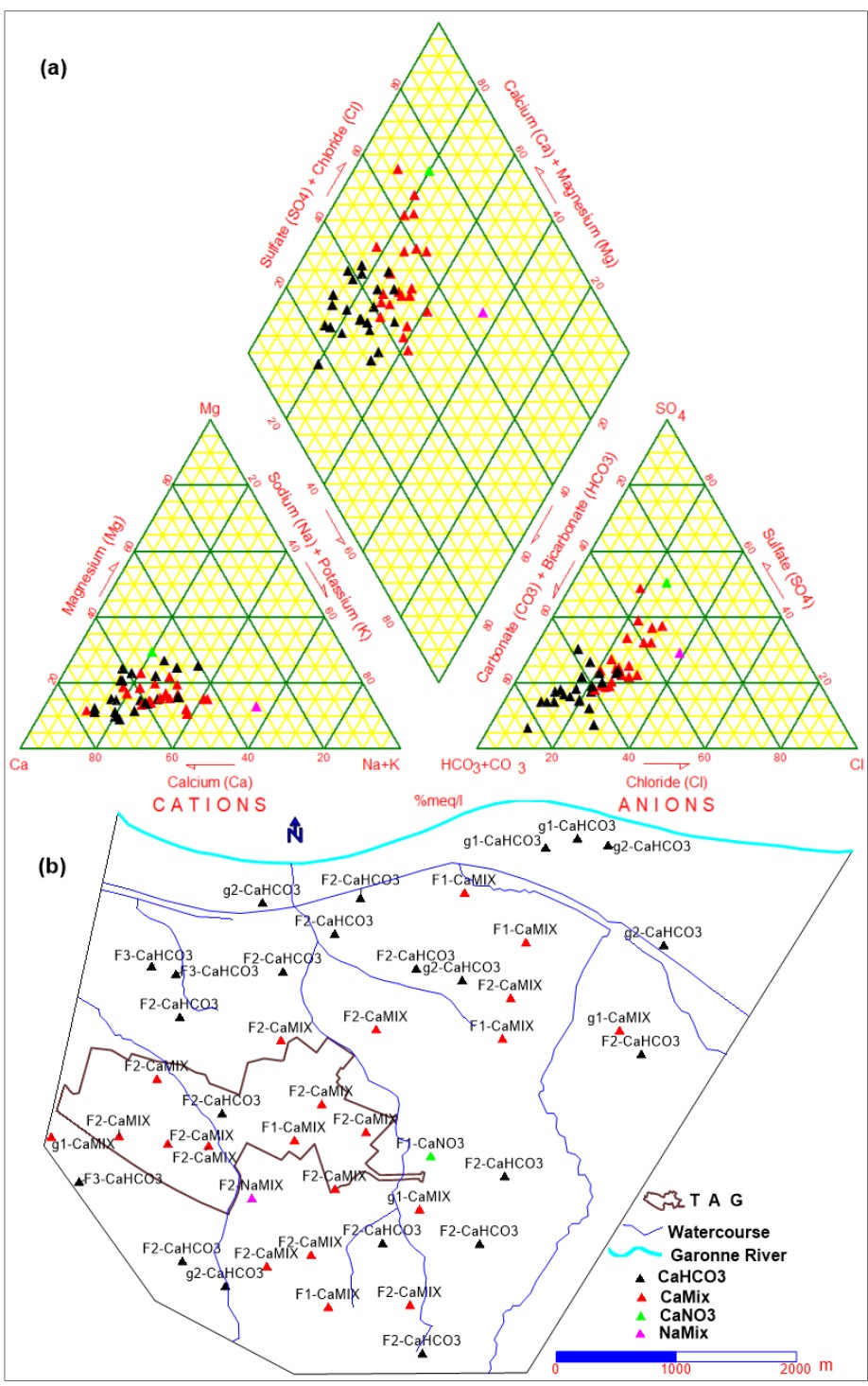

**Figure 11.** (**a**) Modified Piper diagram depicting hydrochemical facies of groundwater in the study area; (**b**) spatial distribution and classification of these facies based on Stuyfzand classification.

The water type changes between Ca-HCO$_3$ and Ca-Mix. The Ca-HCO$_3$ water type is mainly found in the low plain near the Garonne River and in the southern part of the study area, where lateral groundwater inflow can occur along the hillsides. The "Mix" anions water type refers to the water in which no anion family makes up more than 50% of the sum of anions. This water type indicates that more salt dissolution and reactions such as the oxidation of organic matter that increase the HCO$_3^-$ might occur [37]. The local appearance of the water type Ca-NO$_3$, indicates that NO$_3$ is more dominant than the other anions, and strong influence of anthropogenic activities on the water quality is presented.

The Piper trilinear diagram can help to visualize the main hydrochemical facies of groundwater and compare their evolution. The spatial distribution of groundwater samples on the Piper trilinear (Figure 11) confirms the water type classification based on Stuyfzand (Ca-HCO$_3$ and Ca-mixed types). The dominance of Ca water type in the aquifer system might indicate more liberation of Ca$^{2+}$ ions from silicate minerals [38,39]. This water type reveals the influence of the recharge mechanism by the Garonne River or the direct infiltration from the surface and the effects of ion exchange or reversed ion exchange on water quality. The source of HCO$_3^-$ in the groundwater might be due to intense chemical weathering processes in the aquifers or from natural processes such as incongruent dissolution of silicates in the groundwater reacting with carbon dioxide that might release bicarbonates [39,40].

### 5.2.4. Silicate Hydrolysis

The dissolved silica derives primarily from the weathering and subsequent dissolution of silicates and aluminosilicates in bedrocks and soils [41,42]. This process is irreversible, so the silica is retained in an aquatic solution, influenced by thermodynamic factors in the dissolution process, pH, and adsorption of silicate minerals [39,43]. The spatial distribution of silicate values measured in the groundwater collected in September 2018 is presented in Figure 12. The relatively low SIO$_2$ values were mainly measured in the wells near the hillsides and the Garonne River. These values can be explained by a shorter path of the water in the aquifer and by mixing of groundwater with the surface water, respectively. The relatively high values were measured in the central part of the study area, indicating the dissolution processes of aluminosilicate minerals in this area. They might have a longer residence time of the groundwater.

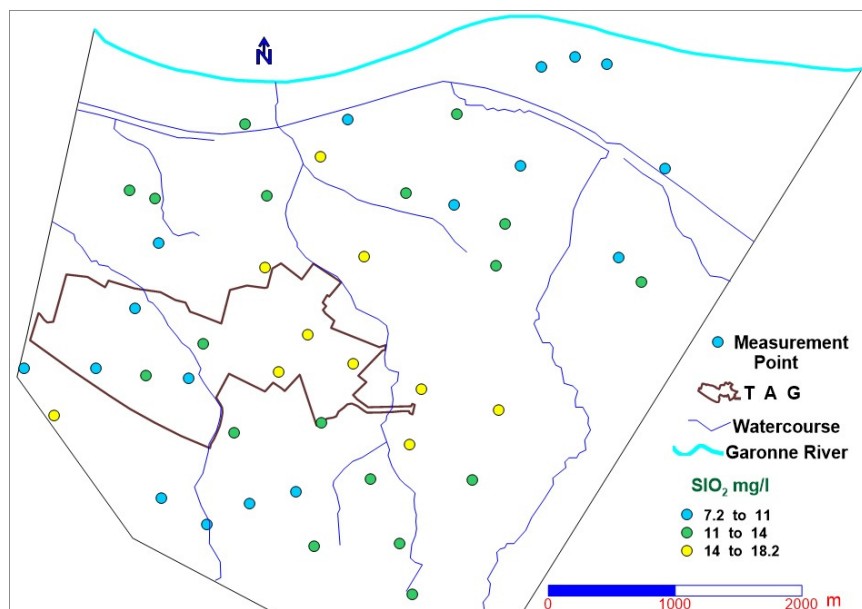

**Figure 12.** Classification and spatial distribution of SiO$_2$ concentration (mgL$^{-1}$) measured in groundwater, September 2018.

The mineral stability diagram is another approach to test the proposed hydrochemical evolution of silicate minerals and their stability [44,45]. These diagrams are usually used to assess the degree of equilibrium and water–rock interaction [46,47]. The relationships between ($[Ca^{2+}]/[H^+]^2$), ($[Na^+]/[H^+]$) and log $[H_4SiO_4]$ is shown in Figure 13. The stability fields for Ca-Plagioclase (anorthite) and its possible weathering products, gibbsite, kaolinite, and Ca-montmorillonite, as a function of log ($[Ca^{2+}]/[H^+]^2$) and log $[H_4SiO_4]$ show that the kaolinite is more likely to be stable than, for example, gibbsite, as a result of anorthite weathering products. However, the stability fields for Na-Plagioclase (albite) and its potential weathering products, paragonite, gibbsite, kaolinite, and pyrophyllite, as a function of log ($[Na^+]/[H^+]$) and log $[H_4SiO_4]$ show that the main reaction is the conversion of albite to kaolinite. The primary silicate minerals, plagioclase and clinopyroxene, can be dissolved and weathered to kaolinite, which is more likely to be in equilibrium in the system.

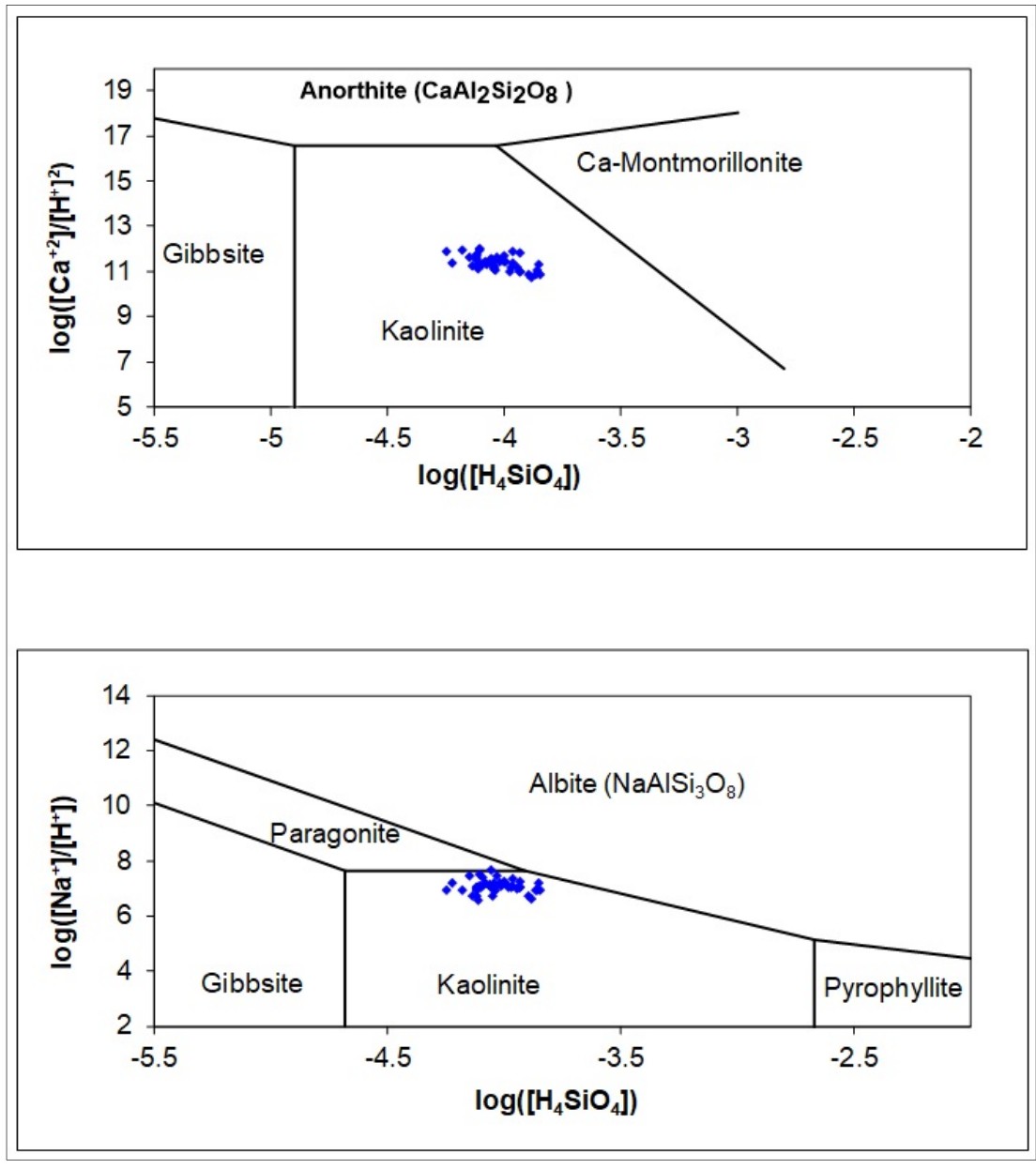

**Figure 13.** Stability diagram of Ca-Plagioclase (anorthite) and Na-Plagioclase (albite) and their possible weathering products.

Nevertheless, the equilibrium between clay and primary silicate minerals is probably the main process controlling groundwater chemistry in the study area. The two diagrams (Figure 13) show that the primary silicate minerals, anorthite and albite, tend to dissolve and transform into secondary minerals in the groundwater system. However, the groundwater is not in equilibrium with either anorthite or albite, and both minerals will decompose into kaolinite, as shown in the following equations:

$$CaAl_2Si_2O_8 + 2H_2CO_3 + H_2O \rightarrow Al_2Si_2O_5(OH)_4 + Ca^{2+} + 2CO_3{}^2 + 2H^+$$

$$2NaAlSi_3O_8 + 2H_2CO_3 + 9H_2O \rightarrow Al_2Si_2O_5(OH)_4 + 2Na^+ + 4H_4SiO_4 + 2HCO_{3-}$$

Ref. [48] have explained the effect of rock dissolution on the groundwater composition through the scattered plots of $Ca^{2+} + mg^{2+}$) versus ($HCO_3{}^- + SO_4{}^{2-}$), as shown in Figure 14.

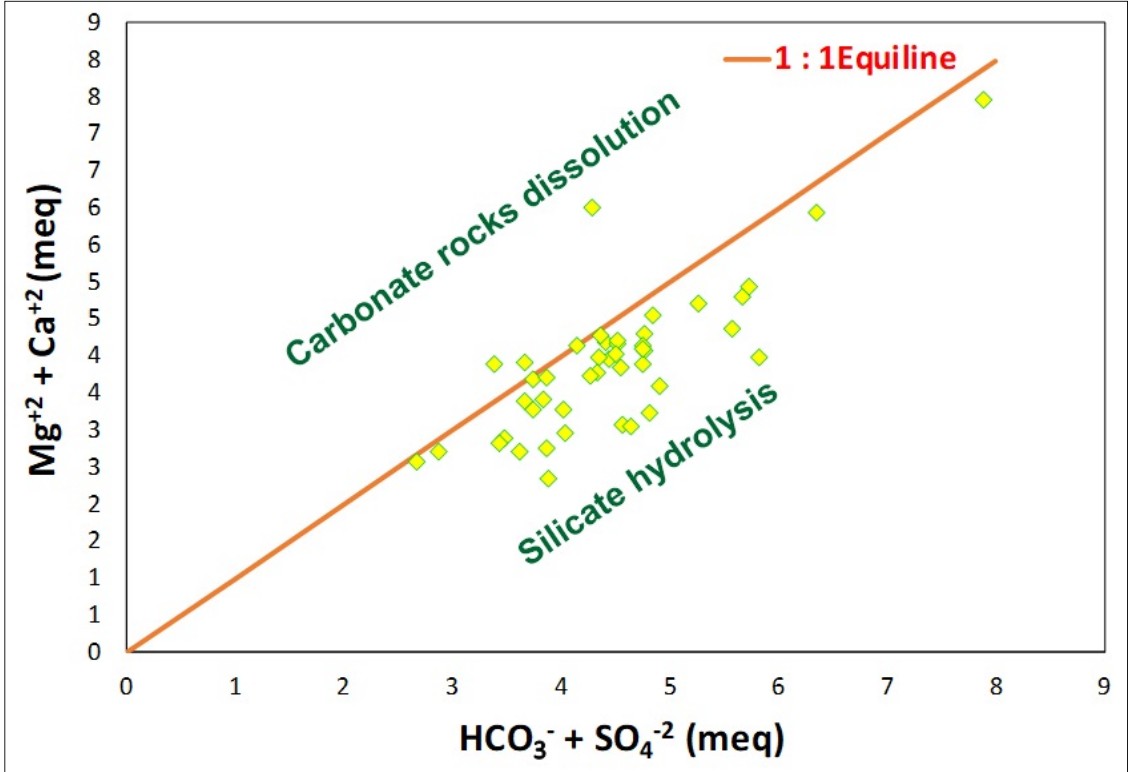

**Figure 14.** Scattered plots of ($Ca^{2+} + Mg^{2+}$) vs. ($HCO_3{}^- + SO_4{}^{2-}$) showing the effect of carbonate rock dissolution and silicate hydrolysis on groundwater composition.

Figure 14 shows that the majority of the water samples are located near or below the equiline on the $HCO_3{}^- + SO_4{}^{2-}$ side, indicating that, on the one hand, silicate hydrolysis is the main hydrochemical process controlling the chemistry of these samples and, on the other hand, the excess $HCO_3{}^- + SO_4{}^{2-}$ can be derived from other processes such as ion exchange.

The effect of the evaporation process on groundwater chemistry was investigated using the molar relation of $Na^+$ and $Cl^-$ (Figure 15). When this ratio is close to one, this indicates that halite dissolution contributes mainly to $Na^+$ concentration in the groundwater. In contrast, most samples have a ratio greater than 1, which signifies silicate weathering or ion exchange [49,50]. In other words, an excess of $Na^+$ in the aquifer system likely results from silicate weathering and ion exchange.

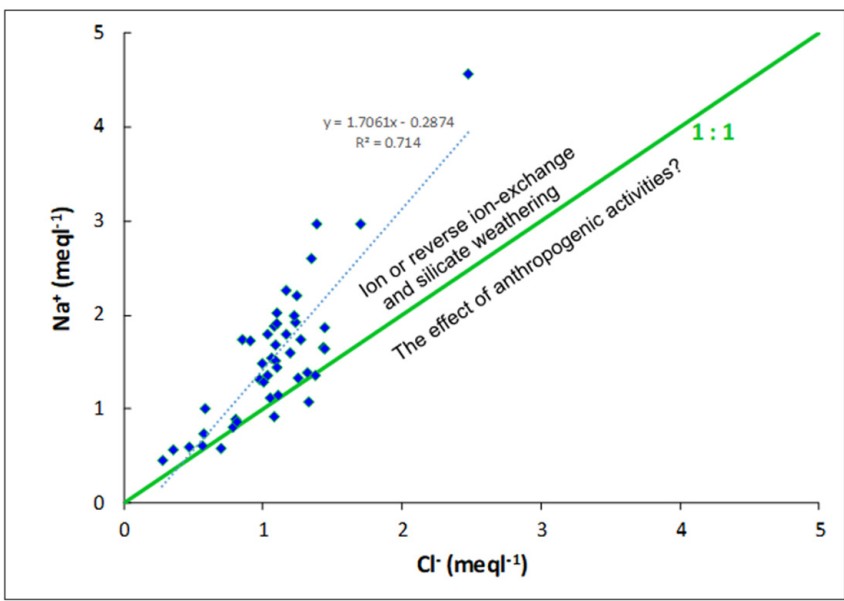

**Figure 15.** Scatter plot of Na$^+$ and Cl$^-$ molar ratio.

### 5.3. Modeling of Groundwater Flow

The model mesh and structure have been generated based on all basic geometric information. The finite element method is characterized by its flexibility in generating the mesh and refining the nodal grid. There are no inactive nodes in the generated mesh because the elements match the model limit. The model domain consists of 17,936 mesh elements and 11,545 mesh nodes (4484 elements per layer and 2309 nodes per slice); the triangulation method was used for mesh generation. A reasonable refinement has been applied to the TAG area to achieve better precision on groundwater flow in this area for potential future infiltration from other retention basins.

#### 5.3.1. Boundary Conditions

Defining suitable model boundary conditions is an essential step in building numerical groundwater models; it is largely responsible for the flow mechanism and the calculation in the system [51,52]. The model boundary conditions were determined according to hydrogeological conditions. The model southern boundary forms the geological limits between the alluvium of the low terrace and the molassic formations of the Aquitanian and Stampian ages, whereas in the north towards the Garonne River, the boundary is naturally established based on the topography and water level of the river in this period. The eastern and western boundaries were assigned as no-flow boundaries since the equipotential contours intersect with these boundaries almost in perpendicular form (Figure 7). The potentiometric map was created based on groundwater levels measurements of March 2019 using the ordinary kriging algorithm method [53]. This was the map for the steady-state model simulation. The south and southwest boundaries upstream of the model area are prescribed constant hydraulic heads (B.C.s) that correspond to manual groundwater level measurements. At the northern boundaries near the river, and since this one flows directly on the substratum molassic, the initial hydraulic heads were taken according to the linear interpolation of the Garonne Riverbed altitude, taking into account the water level of the river during this period. Since the alluvial aquifer overlies an impermeable molassic layer, there is no upward leakage from the model bottom. The direct recharge of the alluvial aquifer by infiltration depends strongly on the rainfall and the permeability of the unsaturated zone. The topsoil cover is globally homogeneous over the model area (vegetal soil, silt, and sandy silt), and a homogeneous annual recharge (70 mmy$^{-1}$) was applied to the first model layer. The lower and upper horizontal model boundaries are limited by the surface of the substratum molassic and the topography of the model top

layer, respectively. According to the water level fluctuations in the wells located near the watercourses in the study area and the piezometric curve shapes, there is no hydrodynamic relationship between the alluvial aquifer and the hydrographic network. The exception is the hydrodynamic relationship between this aquifer and the Garonne River, where the river recharges the aquifer during the flood periods.

### 5.3.2. Hydraulic Conductivity

Hydraulic conductivity is an essential parameter to evaluate the aquifer's potential productivity, response to recharge, and groundwater circulation. The meandering system strongly influences the heterogeneity of the alluvial deposits. The meandering crossing and channel abandonment processes, which allow the conservation of dense clay plugs, are the main cause of sediment heterogeneity in the alluvial plain. As a result, there is significant local variation in the hydraulic conductivity values of the alluvial aquifer. Thus, these values can be changed within a few meters. This variation has strongly affected the productivity of the aquifer from one well to another. This can be explained by the alteration of the alluvial deposits and the presence of paleochannels or clay lenses characterized by low permeability. The hydraulic conductivity values were assigned to the model based on the steady-state trial and error calibration method to simulate the measured hydraulic head and subsequently archive a reasonable agreement between the measured and simulated groundwater levels. These values change between $1.4 \times 10^{-2}$ and $6 \times 10^{-5}$ m.s$^{-1}$, and reflect a large hydraulic heterogeneity even at a small scale in the model area (Figure 16). For the other layers, uniform hydraulic conductivity values were assigned for each layer because of their small effect on the groundwater flow system of the alluvial aquifer.

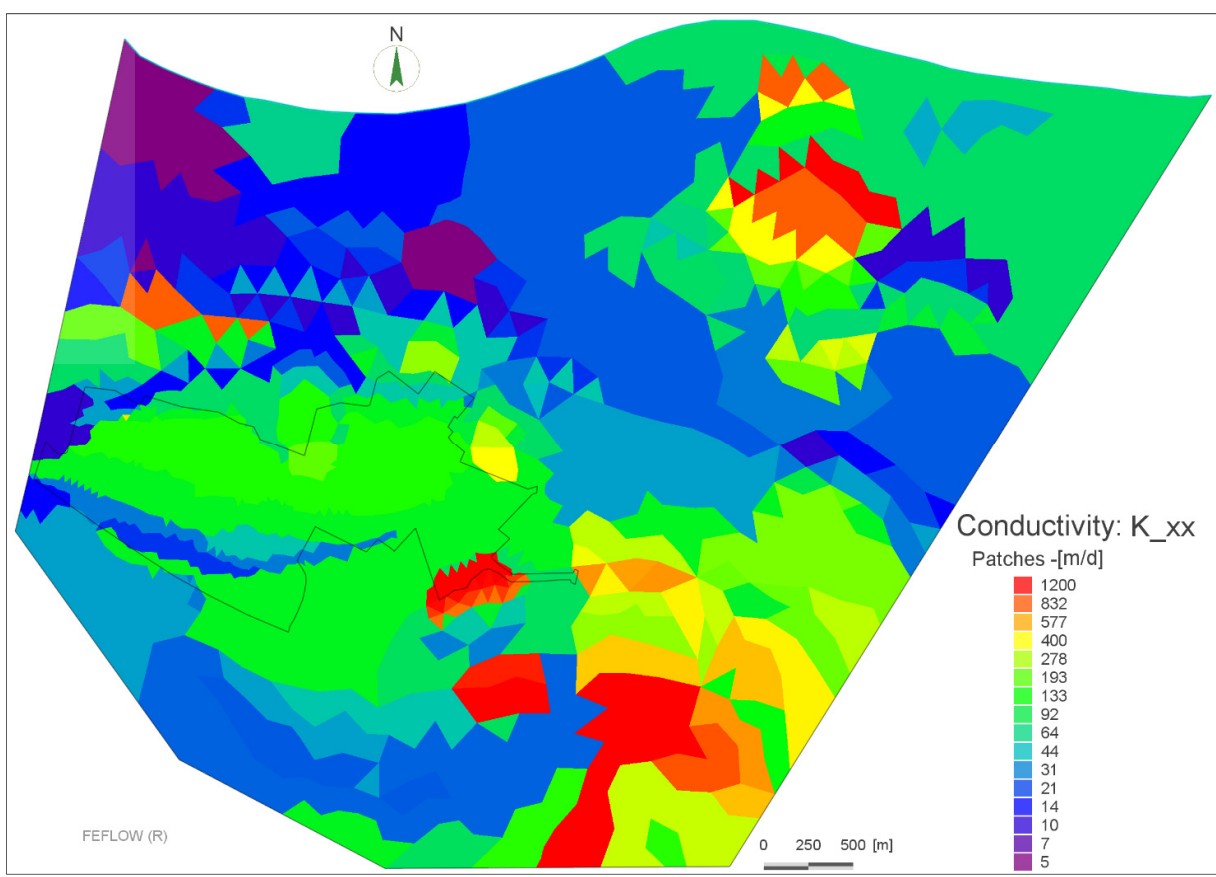

**Figure 16.** A spatial distribution map of the hydraulic conductivity of the alluvial aquifer resulted from the model calibration.

### 5.3.3. Steady-State Calibration

The calibrated model allows us to check the consistency of the data governing the hydraulic performance of the aquifer system, mainly the recharge, permeability, and piezometric levels. The first step in model calibration was to adjust the hydraulic conductivity values using the inverse method in steady-state to simulate the hydraulic heads measured in the field.

The superposition of the measured and calculated groundwater level contours (Figure 17) indicates a good agreement. The calibration is mainly performed by adjusting the hydraulic conductivities to obtain piezometric contours as close as possible to the actual piezometer. The calibrated model defines the initial groundwater level, which is used to run it in the transient regime considering the effect of artificial groundwater recharge from the retention basin.

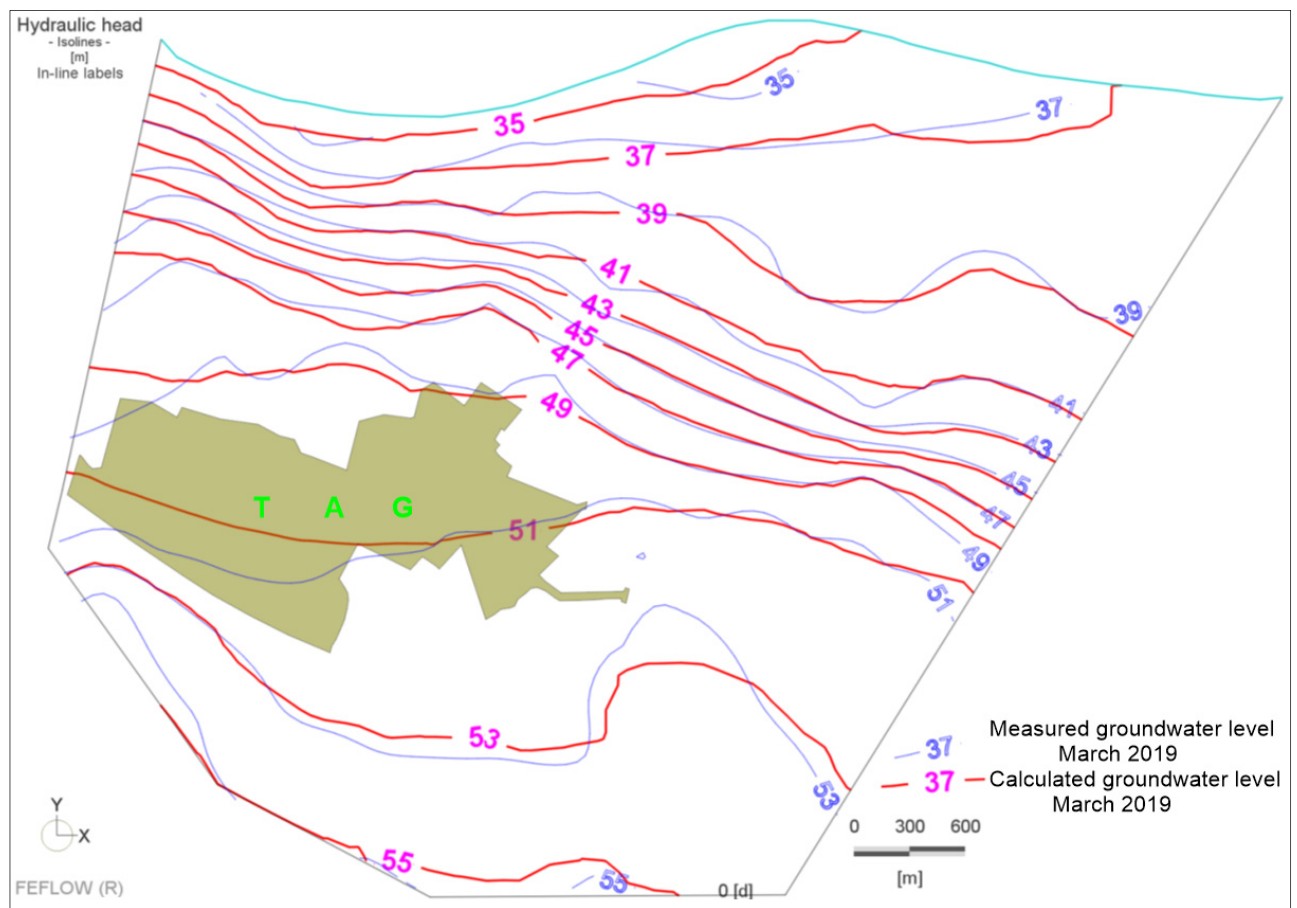

**Figure 17.** Measured and calculated piezometric maps, March 2019.

The scatter plot and regression analysis of the calculated and measured hydraulic heads are shown in Figure 18. The result indicates good agreement between the simulated and measured hydraulic heads.

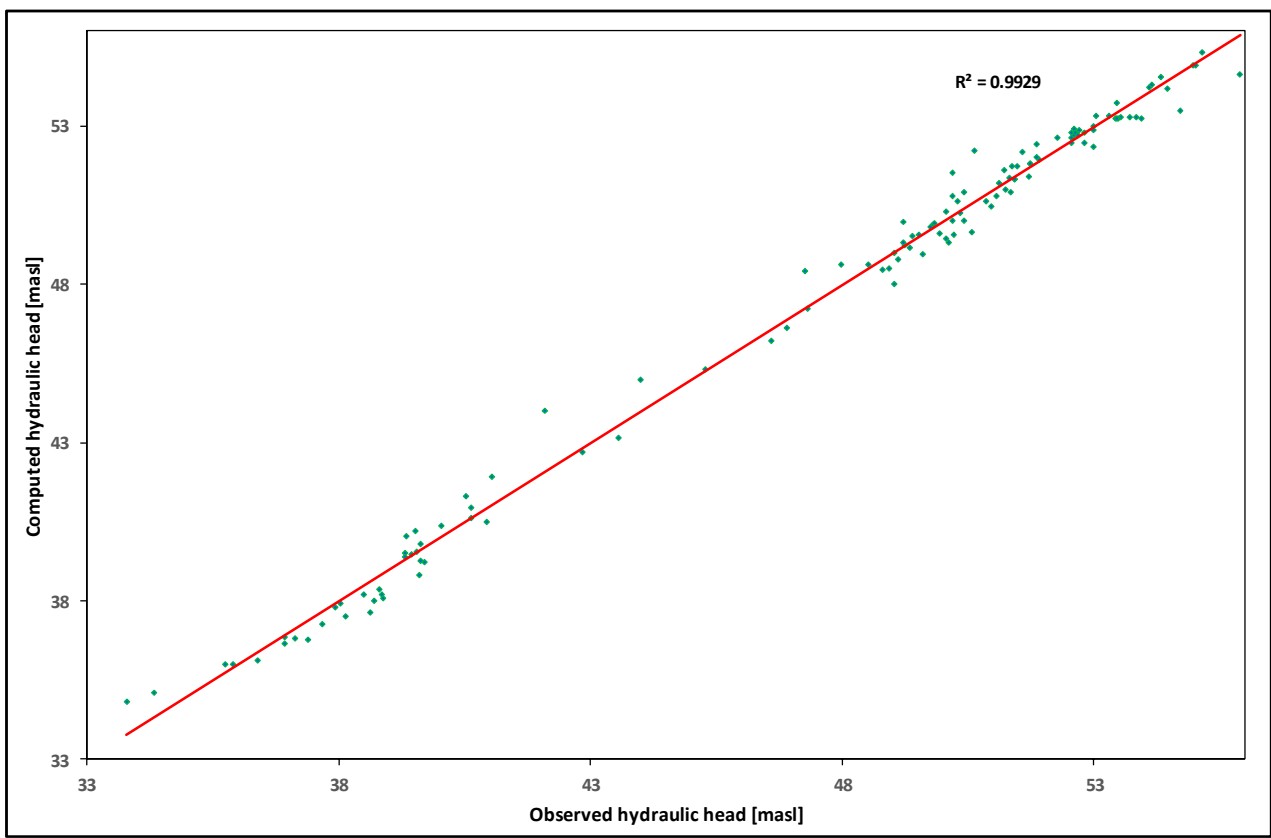

**Figure 18.** Scatter plot of measured and simulated initial hydraulic heads for March 2019.

5.3.4. Results of Transient Modeling

After model calibration, the model was turned into daily time steps to simulate the groundwater flow after the artificial recharge has integrated into the hydrodynamic system of the alluvial aquifer in the TAG area. After sufficient eventual precipitation storm events, the retention basin filled up. Subsequently, the infiltrated water increased the water level in the adjacent wells, as we have measured in two wells near this basin. The calculated equipotential lines show that a piezometric dome was created close to the retention basin after runoff water infiltration.

To simulate groundwater flow patterns after applying groundwater artificial recharge, we selected the nodes corresponding to the infiltration basin in the TAG area in the model mesh. After that, we applied a forward process in a transient state, associated with time series corresponding to groundwater levels measured close to the retention basin, to delineate groundwater path lines in flow direction and to estimate the travel times. From Figures 17 and 19, we can see that the initial equipotential contour located downstream of the retention basin (contour 49 m.a.s.l), moved about 500 m toward a groundwater flow direction after the model was run in the transient state. It was estimated that the infiltrated water would take about four months to reach the Garonne River, which is an appropriate time to sustain its low flow if the recharge occurs between March and April.

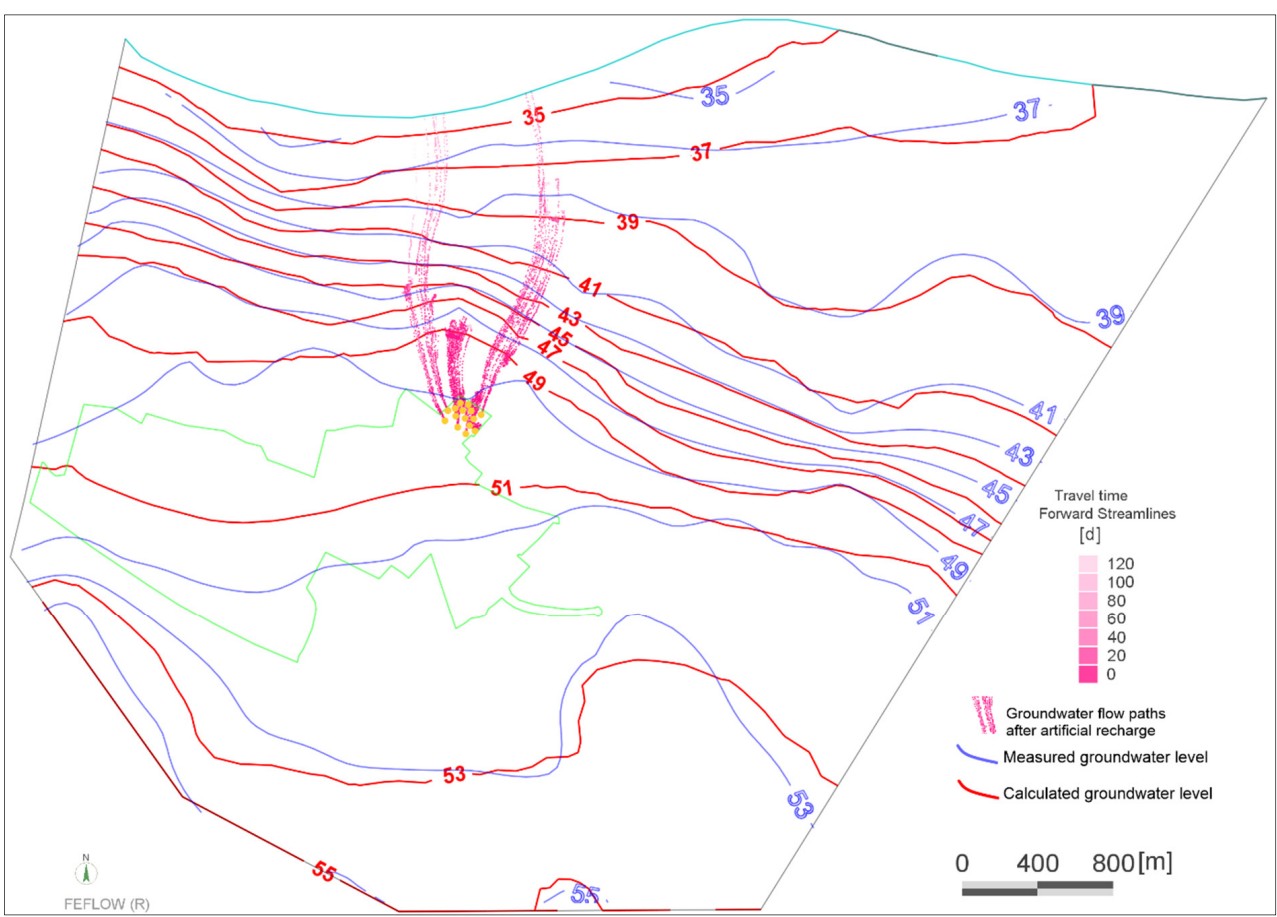

**Figure 19.** Groundwater flow patterns towards the Garonne River after artificial groundwater recharge applied in the TAG area, based on [19].

## 6. Conclusions

Despite the very complex and partially known geological and hydrogeological features of the alluvial aquifer in the Garonne Valley, coupling of hydrochemical characteristics and hydrodynamic modeling supports a better understanding of the mechanisms controlling the water quality and the aquifer dynamic as well as the relationship between this aquifer and the Garonne River. The dynamic exchange between the river and the alluvial aquifer can be considered a major long-term management tool, considering the effects of climate change.

Groundwater level changes show that the variation is greater in the low plain in the northern part near the Garonne River, where the alluvial aquifer drains into the river during the dry periods and is recharged by the river during flood periods. The water levels measured in two wells located near the river indicate the effect of the river on the groundwater level during the flood period. The major drainage axis of the groundwater is generally oriented from south to north toward the Garonne River, which is the main outlet for the aquifer. The piezometric curves show no hydraulic connection between the aquifer and the watercourses in the study area except for the Garonne River.

The nitrate concentrations measured in the alluvial aquifer have confirmed the contamination of this aquifer by nitrates. This contamination is related directly to the strong anthropic pressure, particularly the use of nitrogenous fertilizers in agricultural activities, considering the excess irrigation water and the high intrinsic vulnerability of the alluvial aquifer.

According to the saturation indices values calculated with PHREEQC, the groundwater has negative saturation indices (SI < 0) concerning carbonate minerals (calcite $CaCO_3$ and dolomite $CaMg(CO_3)_2$), as well as evaporitic minerals (gypsum $CaSO_4.2H_2O$, an-

hydrite $CaSO_4$, and halite $NaCl$), indicating that the system is either depleted of such minerals, or the duration of the water–rock interaction (the residence time) is relatively short. However, the dissolution of carbonate rocks is not the major process controlling groundwater chemistry in the study area, and these minerals will continue to dissolve in the groundwater if they are present in the system. In contrast, the quartz saturation index shows a positive value for most of the sample, indicating that the dissolution of aluminosilicate minerals significantly affects the alluvial aquifer's water quality. Hence, silicate hydrolysis is the main hydrochemical process controlling the chemistry of the alluvial aquifer in the study area.

The groundwater samples fall within the $Ca$-$HCO_3$ and Ca-mixed types, and just two samples have shown $Ca$-$NO_3$ and Na-Mix types according to Stuyfzand classification. Silicate weathering and ion exchange are the main processes controlling groundwater chemistry in the study area.

The groundwater level changes measured in two boreholes near the retention basin show groundwater levels increasing between 0.5 and 1 m after a storm and runoff water infiltration. The modeling results show that the infiltrated water would take about four months to reach the Garonne River, which is an appropriate time to sustain its low flow if the recharge occurs in spring. This result shows that the alluvial aquifer in the study area is potentially interesting for the groundwater artificial recharge to maintain the groundwater level and sustain the low flow of the Garonne River. This strategy can contribute to ensuring the proper function of the river environment and creating humid zones in the alluvial Garonne Valley. This pragmatic and potentially eco-responsible management tool can be considered in the context of sustainable development as an adaptation strategy to climate change in the region. Excess water generated by the flood or snowmelt in the Garonne basin, which is characterized by good quality, could be used for artificial recharge of the shallow alluvial aquifer, well connected with the river, once suitable sites have been identified.

Future work should focus on obtaining more data about the aquifer's hydraulic conductivity, flood event, groundwater temperature measurement, groundwater extraction for agricultural use, and other water consumption in the area. This information can be useful for groundwater hydrodynamic and thermal modeling and to calculate the study area's water budget, as well as to show the effect of artificial recharge on the river discharge. More groundwater sampling before and after the recharge must be considered to specify the qualitative effect. Thus, this work will involve measuring the thermal plume induced by recharged water within the aquifer. These measurements will also be modelled in a coupled hydrodynamic and thermal model in order to simulate the aquifer's buffering effect on temperature. The coupled model will also serve as a prospective tool for quantifying the effect of recharge under warmer atmospheric conditions.

**Author Contributions:** N.A. performed field activities and wrote the original draft of the paper. N.A., A.D., P.M. and M.F. designed the methodological framework. N.A. and A.D. performed the simulation and analyzed the results. All co-authors collaborated on the redaction, review, and editing of the manuscript. All authors have read and agreed to the published version of the manuscript.

**Funding:** This research has been supported by the Nouvelle Aquitaine Region (File no 2017-1R20104) and by SMEAG, Syndicat mixte d'études et d'aménagement de la Garonne (File no CT_2019-068).

**Institutional Review Board Statement:** The study was conducted in accordance with the Declaration of Helsinki, and approved by the Institutional Review Board.

**Informed Consent Statement:** Not applicable.

**Data Availability Statement:** Data is unavailable due to privacy restrictions.

**Conflicts of Interest:** The authors declare no conflict of interest.

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
