# Peer review of "Hydro-Geochemical Characteristics of the Shallow Alluvial Aquifer and Its Potential Artificial Recharge to Sustain the Low Flow of the Garonne River"

_water, doi:10.3390/w15162972_

Round 1

Reviewer 1 Report

- In the abstract, try to use some value (water level fluctuation, quality+++)

- More detail, the oscillation of water level vs natural and anthropogenic activities

- What about the tectonic in this intercommunication?

- using the groundwater level map means The piezometric map??? if yes modify your text

- What about the hydrodynamic parameters of your aquifer ???

- More detail on the budget for the 2 periods (recharge and discharge)

- Added the value of the TDS, and detailed the quality of groundwater vs transit time

Needed minor revision

Author Response

Please kindly check the authors' response in the attachment.

Reviewer 2 Report

In the article “Hydro-geochemical Characteristics of the Shallow Alluvial Aquifer and its Potential Artificial Recharge to Sustain the Low Flow of the Garonne River” an interesting case study considering manage aquifer research (MAR) is presented. Particularly interesting is the aim of the aquifer recharge to maintain ecological flow of a river during dry periods. 

However, there are several major problems with the article:

1. The overall quality of writing is poor, at places it is difficult to flow the story line or to understand the text. At places there are duplications or in a single paragraph several contrasting ideas are presented, that impairing the readability of the article. The flow of narrative should be improved.

2. Three main groups of the results are presented: groundwater level measurements, hydrological model of the study area and hadrochemical data. The groundwater level measurement data supports and is integrated into development of the hydrological model thus forming integral part of the same study. But the hadrochemical data stands apart and is of little use for understanding or interpreting the modeling results. The authors should either incorporate the results of hydro chemical analysis in discussing the modeling results or omit this section of the article.

3. The article is lacking discussion section at all – the presentation of the results abruptly ends with the conclusions. In fact the main finding of the article is justified by just one sentence “It was estimated that the infiltrated water would take about four months to reach the Garonne River…..”. I find no description of the methods of how this was estimated, no discussions on the uncertainty of such a claim or estimation of how many managed aquifer recharge events are likely per year. Furthermore, in the caption for the Figure 17 we see a reference to the conference abstract by the same authors. For the article to be published authors need to elaborate on the methodology, uncertainty and implication for this crucial finding. In addition the thermal effects of the managed aquifer recharge is not discussed at all in the results and methods sections – how can you draw any conclusions regarding the groundwater temperature if you do not investigate that?

4. To my understanding, a crucial part of the article should be discussing the volumes of the aquifer recharge in TAG project and estimation how many similar projects in the Garonne river basin should be implemented so that a meaningful impact on the river discharge is obtained. 

5. As a geologist I stress that the authors have to present a conceptual hydrogeological cross section of the study site, clearly identifying the relationship between different types of alluvium, the molase sediments and other geological units. They can find guidance for conceptual hydrogeological modeling in this document: https://voda.hr/sites/default/files/dokumenti/propisi-i-obrasci/26_-_guidance_on_risk_assessment_and_the_use_of_conceptual_models_for_groundwater_-_eng.pdf (Technical Report - 2010 - 042 Common implementation strategy for the Water Framework Directive (2000/60/EC) Guidance document No. 26 Guidance on risk assessment and the use of conceptual models for groundwater)

6. Another crucial remark – a short but informative description of the TAG project should be provided: when and by whom was it established? Is it part of a large programmed? what is the drainage catchment? what is the area of the infiltration ponds? Any other crucial details?

Apart from these major remarks I have several minor comments:

Line 16 – please indicate the region and country

Line 28 – please add minimum background information about the TAG project.

Line 41 – I suggest adding keyword MAR – managed aquifer recharge

Line 48 – add the information about study region and name of the country

Line 84 “one objective” – if there are other objectives, please state them, if there are no other objectives the remove “one”

Line 84-93 – I suggest that this should be a new paragraph

Line 94-101 – more details about TAG project is needed (see above)

Line 99 – a reference is needed regarding the water temperature. The sentence should be reformulated, it is difficult to understand

Line 114 – please start the paragraph with more general information, like the location and altitude and the move on in describing local details of the study site/ 

Line 127 – the paragraph should de rewritten describing the geological setting instead of geological events in the past – from the point of view of the hydrology we are interested on the actual relationships between different structures rather than how they arise. In addition pleas add description of the mineral composition of the different geological units, as this is needed as background information for the interpretation of the hydro chemical data.

Line 150-152 – sentence “The hydrological…” this is the case for ALL rivers, omit the sentence

Line 161 – please describe HOW the climate change is impacting the river. In addition information is needed how the low flow period is defined? What is the minimum discharge target? How the mountain reservoirs are managed?

Line 171 – Figure 3 – graph need more explanation – what are the scattered dots? Why the vertical axis is months? In addition English should be used in the graph.

Line 174 – the paragraph should be rewritten systematically describing the geological, geomorphological and hydrological features. A conceptual hydrogeological cross section would greatly help

Line 191 – “underground part..” – aquifers by definition are located under the ground surface – think of better wording

Line 202 – “measure” –  describe how the measurements were done? Where new piezometers installed or existing wells used? 

Line 203 – Figure 7 – please correct the figure numbering according the order they are referred in the text

Line 216-218 – pleas expand the description of the TAG and move it to the beginning of the site description – first present background information and then flow by specific details.

Line 218 – “green and moderate industrial zone” – what is that? Please explain.

Lines 218-221 – please explain how the water table monitoring was donw: where? How? How long? Why? Who?

Line 244 – Figure 6 – the core section is not informative at all – either remove it or improve resolution and readability

Line 246 – explain – what is slice?

Line 259, Line 219 – pleas, clarify, what was the duration of the groundwater level monitoring

Line 257 – start a new a paragraph here

Line 289 – pleas alinge the horizontal scales of the two groundwater level graphs

Line 299 – adjust the subchapter name to “nitrates” or similar

Line 300-308 – this part should be moved to introduction

Line 332 – add Mg and Ca concentrations into one number so that it matches the units of PHREEQC simulation. Consider using different shapes for different parameters

Line 347 – “g and F” – explain what does it means, for a reader that is not familiar with classification you are using

Line 351 – “between 1, 2 and 3” – explain, what is this?

Line 356 – “salt dissolution” – name what could realistically be these salts in your location

Line 368 and 369 – dissolution and weathering in this context is the same 

Line 402-407 – aquifer mineralogy need to be described for context either here or better in the site description

Line 430 – “scootered” – what is this?

Line 433 – 441 this should be moved to the introduction or site description

Line 452-480 – I suggest that you systematically rewrite this paragraph by improving the structure, for example by describing the model boundary and the stating the type of boundary condition you apply there. In addition this is description of a method and should be moved to the methods section.

Line 491 – “few meters of magnitude” – what is that? Reformulate.

Line 482-503 – a lot of duplications in the paragraph – rewrite please.

Line 519 – figure 17 – considered removing the mesh and grey background color to improve the readability of the image

Line 508 – the calibration procedure of the model should be described in the methods section

Line 536-537 – this is the essence of your publication, but it is not supported by description of method how do you estimate this (except the figure 19, that lacks explanation) I guess that the temperature buffering hides behind this sentence as well, but it need to be explained explicitly. Please, add estimations of the water volumes involved and put them into wider context of overall river discharge and impact this project have.

Line 539 – figure 19 – this is the most important figure of your publication. Considered improving its readability by removing or fainting the model mesh and explaining the flow tracing in the image caption as well as manuscript text, both methods and results sections. 

Use of the "However, .." in many places seems to be inappropriate in many cases. It should introduce a sentence that contradicts the previous statement.

Author Response

(The authors gave the same response as above.)

Reviewer 3 Report

*Please note that this report should be : Accept after minor revision (corrections to minor methodological errors and text editing)*

·        Abstract is too lengthy, it can be rewritten into brief to reflect the entire paper

·        Too many figures, the present form looks like a report, authors should think reduce the number of figures, include only required and important

Otherwise, paper is nicely discussed hydrochemical process and groundwater modeling in part of Garonne River  which could be interest of wider auidence. Hece, I recommend for publication after incorporaton of the above suggestions.

Author Response

(The authors gave the same response as above.)

Round 2

Reviewer 1 Report

It can be accepted in this form

Reviewer 2 Report

The manuscript has been improved, and most of the issues have been addressed.

One final remark:

Figure 10 – I suggest that you sum the measured Ca and Mg (Ca + Mg) molar concentrations so that it can be directly compared to PHREEQC simulation. Furthermore, expressing the Ca and Mg concentrations equivalents so that thy can be directly compared to the HCO3 concentrations.